# Mechanism of trinucleotide repeat expansion by MutSβ-MutLγ and contraction by FAN1

Issam Senoussi[1,2], Valentina Mengoli[1], Arianna Cerana[3], Andrea Rinaldi[3], Andrés Marco[4], Giordano Reginato [1], Simone G. Moro [1], Ananya Acharya[1], Megha Roy [1], Akshay Jayachandran [1], Elda Cannavo[1], Ilaria Ceppi [1] & Petr Cejka [1] ✉

Triplet repeat expansion underlies multiple pathologies, including Huntington's disease, often arising in somatic non-dividing tissues such as the brain. Despite identification of genetic modifiers, mechanistic insights remain limited. Using purified human proteins, we show that MutLγ (MLH1-MLH3), stimulated by MutSβ (MSH2-MSH3), incises DNA opposite an extrahelical loop on the 5' side. This activity, with a moderate sequence preference, generates DNA nicks enabling Polδ-mediated displacement synthesis with the loop as a template, leading to expansion. PCNA confines these MutLγ incisions near the loop. FAN1, instead, preferentially targets the looped strand. RFC-PCNA stimulate and direct FAN1 nuclease to the 3' boundary of the loop while restricting its exonuclease activity. No pre-existing nick is required. Following FAN1-RFC-PCNA action, Polδ removes the loop and resynthesizes DNA, causing contraction. FAN1 also directly inhibits MutLγ, preventing its activation by MutSβ. Our study illuminates both repeat expansion and contraction mechanisms and reveals the protective function of FAN1.

Repeat expansion disorders result from the lengthening of tandem arrays of repetitive DNA sequences[1–3]. Huntington's disease, Fragile X syndrome, Myotonic dystrophy and Friedreich ataxia are among more than 50 diverse pathologies linked to abnormal expansions of specific trinucleotide repeats. While repeat lengthening can take place in the germline during meiosis, a characteristic feature of these pathologies is expansion occurring in somatic and particularly non-dividing tissues such as the brain[1,2]. To date, numerous reports based on various model systems or human genome-wide association studies identified key genetic modifiers of these disorders, particularly in conjunction with Huntington's disease, linked to the expansion of CAG repeats in the *HTT* gene[4–8].

Repeat expansion is driven by a subset of factors related to postreplicative mismatch repair. In canonical mismatch repair, base-

base mismatches or small insertion-deletion loops are identified by MutSα (MSH2-MSH6), while small and larger loops are bound by MutSβ (MSH2-MSH3). The recognition of the anomaly by MutSα or MutSβ triggers the recruitment of the MutLα nuclease (MLH1-PMS2), with PMS2 being the catalytic subunit. PCNA, loaded by RFC at strand discontinuities, activates and directs the nuclease activity of MutLα, leading to additional nicks in the discontinuous strand, which typically carries the incorrect DNA sequence[9–11]. The incised strand is subsequently degraded and resynthesized, leading to the restoration of the correct sequence. The mismatch repair pathway reduces mutation rates of DNA replication by up to three orders of magnitude, and hence significantly contributes to genome stability. Paradoxically, a subset of mismatch repair proteins acts pathologically to promote triplet repeat

[1]Faculty of Biomedical Sciences, Institute for Research in Biomedicine, Università della Svizzera italiana (USI), Bellinzona, Switzerland. [2]Department of Biology, Institute of Biochemistry, Eidgenössische Technische Hochschule (ETH), Zürich, Switzerland. [3]Faculty of Biomedical Sciences, Institute of Oncology Research, Università della Svizzera italiana (USI), Bellinzona, Switzerland. [4]Data Curators B.V, DH Oegstgeest, The Netherlands. ✉ e-mail: petr.cejka@irb.usi.ch

expansion. MutSβ plays a critical role in triplet repeat expansion, while MutSα is not involved[12]. There has been a debate about the role of the MutL homologs. Most reports suggest that MutLα does not promote triplet repeat expansion, and that expansion is driven by MutLγ (MLH1-MLH3), and its nuclease activity[13–17].

MutLγ has only a minor, if any, role in mismatch repair[18,19], and its most prominent physiological function is in meiotic recombination[20–22]. The MutLγ nuclease, in conjunction with meiotic MutSγ (MSH4-MSH5), was proposed to nick meiotic recombination intermediates in a way that favors exchanges, termed crossovers, between recombining chromosomes. This activity generates diversity and helps separate recombining chromosomes. PCNA was identified as the likely factor that drives the directional incision by MutLγ, drawing parallels between the regulation of MutLα in mismatch repair and MutLγ in meiotic recombination[23,24].

Mechanisms that govern triplet repeat instability are not well understood. Previously, it was demonstrated that MutSβ promoted DNA cutting by MutLγ on a DNA strand opposite an extrahelical loop[25]. How this activity results in repeat expansion, and how it is regulated, has not been clarified. One of the key factors that prevents repeat expansion is FAN1. FAN1 was initially identified and further characterized based on its physical interaction with MLH1, and its role as a structure specific nuclease in DNA crosslink repair[26–32]. Its protective function against triplet expansion is functionally separate, but still dependent on its nuclease activity[17,33–36]. FAN1 was additionally proposed to disrupt the interaction between MutSβ and MutLγ[33]. The FAN1 nuclease activity was found to be activated by PCNA loaded at nicks on the looped DNA strand, resulting in additional cuts to the looped strand[37]. A source of the pre-existing nicks in a physiological context in non-replicating cells is not clear. Intriguingly, FAN1, together with additional unknown factors, was found to possess an activity leading to loop excision in human cell extracts[37].

Here, we identified the minimal components and mechanism leading to repeat expansion, independent of DNA breaks and replication. The process involves MutSβ, MutLγ, RFC, PCNA, RPA and Polδ, in agreement with recent Huntington's disease mouse and human cellular models, and genome-wide association studies[7,13–17]. We also show how FAN1, through both nuclease-dependent and independent functions, prevents pathological DNA incisions by MutLγ and hence inhibits repeat expansion. Notably, we identified a mechanism involving FAN1, RFC, PCNA and the exonuclease activity of Polδ that can catalyze repeat contraction. Our findings reveal mechanisms underlying trinucleotide repeat instability, which explain numerous genetic studies and highlight processes amenable to therapeutic interventions.

## Results

### EXO1 and RFC-PCNA do not stimulate DNA incisions by MutSβ and MutLγ

The nuclease activity of the heterodimer pairs MutSγ (MSH4-MSH5) and MutLγ required for meiotic recombination is promoted by RFC-loaded PCNA and by EXO1, as shown previously (Fig. 1a–d and Supplementary Fig. 1a, the nuclease-deficient EXO1 D173A variant was used)[22–24,38,39]. In contrast, when the same MutLγ nuclease was assayed together with MutSβ, RFC-PCNA and EXO1, D173A did not stimulate DNA nicking, and instead moderate inhibition was observed (Fig. 1a–d and Supplementary Fig. 1a). The MutLγ nuclease is therefore regulated differently depending on whether the MutS heterodimer partner is MutSβ or MutSγ. The nuclease activity of MutSβ-MutLγ requires ATP, but not ATP hydrolysis (Supplementary Fig. 1b). The observed DNA cleavage was dependent on the nuclease activity of MutLγ, as reactions with a MutLγ variant bearing three substitution mutations in the nuclease active site of MLH3 did not show any activity (MutLγ 3ND, D1223N, Q1224K and E1229K) (Supplementary Fig. 1b, c). We note that these non-specific nicking assays were carried out with negatively supercoiled DNA that permits the loading of PCNA[40]. We found that a

small proportion of the DNA substrate was linearized (Supplementary Fig. 1b, c), corresponding to the capacity of MutLγ to incise DNA opposite nick sites, in accord with previous observations with the yeast homolog[41].

### MutSβ-MutLγ cleave DNA opposite extrahelical loops and are inhibited by RFC-PCNA and EXO1

To gain insights into the positions and regulation of DNA incisions by MutSβ-MutLγ, we constructed a variety of plasmid-based DNA substrates with mispairs, extrahelical loops and/or nicks at various positions (Supplementary Fig. 2a, b). We then analyzed the nuclease reaction products by Southern blotting with radiolabeled strand-specific probes. To verify our methodology, we first confirmed that the canonical mismatch repair proteins MutSα and MutLα are activated by a G/T mispair and directed by PCNA loaded at a nick site (Supplementary Fig. 3a–e). In accord with the seminal work by Modrich and colleagues[9], we observed efficient DNA cleavage by MutLα that was uniformly distributed on both sides of the G/T mismatch, with a strong preference toward the nicked DNA strand (Supplementary Fig. 3a–e).

We next carried out assays with recombinant MutSβ-MutLγ and a DNA substrate harboring an extrahelical $(T)_4$ loop. As shown previously[25], MutSβ-MutLγ did not notably cut the looped DNA strand (less than 10% DNA degradation observed with probes P1 and P2), while the strand opposite the loop was cleaved efficiently (up to 80% DNA degradation detected by probe P4) (Fig. 1e–i). Several additional important conclusions could be made from these experiments. RFC-loaded PCNA had a moderate inhibitory effect on DNA cleavage as scored by overall substrate utilization, while it had a major impact on the positions of the DNA cleavage sites (Fig. 1g–i). Without PCNA, we observed nicking at multiple locations, while in the presence of PCNA, DNA cleavage was restricted to sites closer to the loop, but always preferentially on the DNA strand opposite the loop (Fig. 1g–i). The non-uniform distribution of MutLγ incision sites was in sharp contrast to the reactions with MutLα, which produced a uniform pattern (compare Supplementary Fig. 3a, lane 4, Supplementary Fig. 3b, lane 3 and Fig. 1g, lanes 3 and 4). The preferential incision of the DNA strand opposite extrahelical loops by MutSβ-MutLγ was observed with both $(T)_4$ and $(CAG)_4$ extrahelical loops (Supplementary Fig. 3f–i, lanes 5) without strand discontinuities. In contrast, the combination of MutSβ-MutLα was stimulated by RFC-PCNA and weakly cleaved both strands, with a moderate preference towards the looped strand (Supplementary Fig. 3f–i, compare lanes 3 and 4), likely mediated by largely random orientation of PCNA loaded at loop sites[40,42]. Instead, the absence of a nick in the DNA substrate had no effect on the efficacy nor on the DNA cleavage positions by MutSβ-MutLγ, although the inhibitory effect of PCNA was diminished without a nick (Supplementary Fig. 4a, b). The observed DNA incisions are dependent on the intrinsic nuclease activity of MutLγ (Supplementary Fig. 4a).

EXO1 D173A was found to be moderately inhibitory when combined with MutSβ-MutLγ (Supplementary Fig. 4a, b), in agreement with its protective function in triplet repeat expansion[43,44] and our nicking assays (Fig. 1d). To dissect how EXO1 inhibits MutSβ-MutLγ, we performed nicking assays with a panel of EXO1 variants: the nuclease-inactive D173A, D173A combined with MIP mutations (F506A/F507A) and with I403E that disrupt MLH1 binding[39], and D173A combined with K185D/K237D mutations that weaken DNA binding[39,45]. Both the D173A and the D173A/MIP/I403E proteins inhibited MutSβ-MutLγ cleavage similarly, whereas the D173A/K185D/K237D mutant inhibited MutSβ-MutLγ to a lesser extent (Supplementary Fig. 4c). These data demonstrate that EXO1's ability to impede MutSβ-MutLγ depends primarily on its DNA-binding function rather than its interaction with MLH1 and may result from a non-specific competition for DNA. These results stand in contrast with a critical role of EXO1 to promote MutSγ-MutLγ and its role in meiotic recombination[23,24,38,39]. Finally, in contrast to a

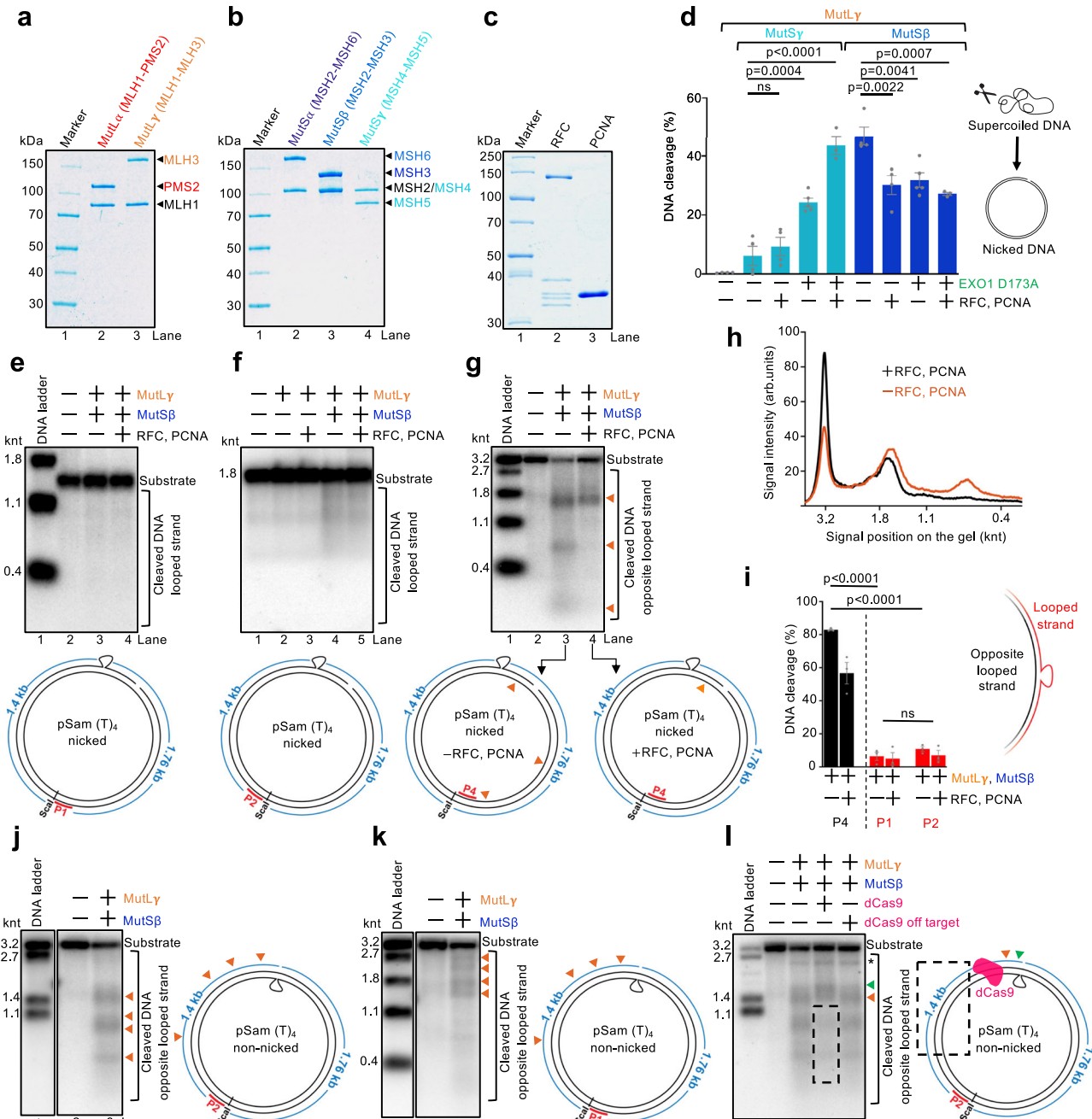

**Fig. 1 | MutSβ-MutLγ cleave DNA opposite to extrahelical loops on the 5′ side.**
**a–c** A representative of three polyacrylamide gels was stained with Coomassie Brilliant blue and shows recombinant proteins used in this study. **d** Quantification of nicking assays with 5.6 kb-long supercoiled DNA (scDNA) and the indicated proteins. See Supplementary Fig. 1a for more details. Averages shown; error bars, s.e.m.; $n = 4$ independent experiments except for bars 5 and 9 ($n = 3$), and for bars 4 and 8 ($n = 5$). Statistical analysis was performed by ordinary one-way ANOVA with Tukey's multiple-comparisons test. ns, non-significant. Nuclease-deficient EXO1 D173A contains the point mutation D173A. (**e–g**), Top, representative nuclease assays with nicked pSam_(T)$_4$ (loop in the top strand) and the indicated proteins analyzed by Southern blotting with probes complementary to the looped strand, P1 (**e**), P2 (**f**), or to the strand opposite the loop, P4 (**g**). Bottom, a schematic of the DNA substrate with indicated probes. The orange triangles denote approximate DNA cleavage positions. **h** Densitometric profile of lanes 3 and 4 of (**g**). RFC and PCNA restrict DNA cleavage by MutSβ-MutLγ. **i** Quantification of Southern blot-based

nuclease assays, such as shown in (**e–g**). Statistical significance was determined by two-way ANOVA followed by Tukey's multiple-comparisons test comparing MutLγ and MutSβ cleavage on the opposite looped strand versus the looped strand. Averages shown; error bars, s.e.m.; $n = 3$ independent experiments. **j, k** Left, representative of three nuclease assays with pSam_(T)$_4$ (loop in the bottom strand) and the indicated proteins. The reaction products were analyzed by Southern blotting with probes complementary to the strand opposite the loop at various positions with respect to the ScaI DNA cleavage site, P2 (**j**) or P1 (**k**). Right, a schematic of the respective DNA substrates with indicated probes. The orange triangles denote approximate DNA cleavage positions. **l** Experiments as in (**j**), from three independent repeats, but with catalytically-dead Cas9 (dCas9) where indicated. The dashed rectangle indicates a DNA region protected from MutLγ cleavage by dCas9. The green triangle indicates additional DNA incisions between the loop and dCas9, caused by the accumulation of MutLγ. *, DNA not cleaved by ScaI. Source data are provided as a Source Data file.

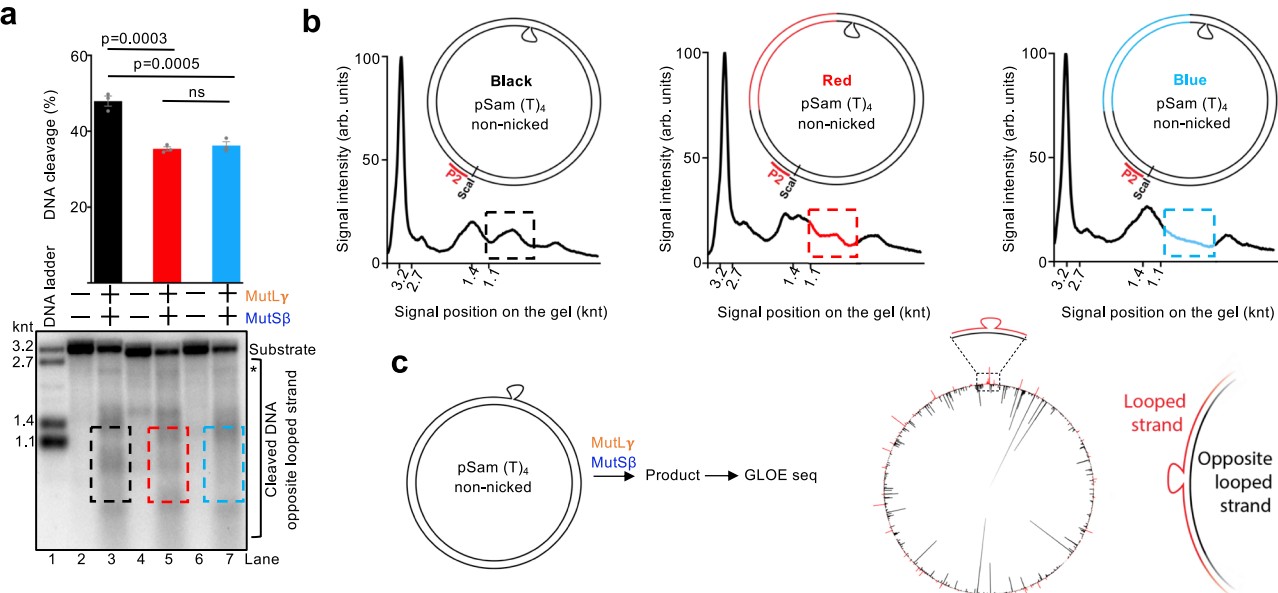

**Fig. 2 | MutSβ-MutLγ recognize diverse loops and cut DNA with a moderate sequence preference. a** Nuclease assays with pSam_(T)₄ or its derivatives (loop in the bottom strand) and the indicated proteins analyzed by Southern blotting with a probe complementary to the opposite looped strand, P2. Where indicated, a segment of the standard "Black" sequence of pSam was replaced with different sequences of the same length, denoted as "Red" or "Blue". Shown are representative assays. Averages shown; error bars, s.e.m.; $n = 3$ independent experiments. Statistical analysis was performed by ordinary one-way ANOVA with Tukey's multiple-comparisons test. ns, non-significant. *, DNA not cleaved by ScaI.

**b** Densitometric profile of lanes 3, 5 and 7 of (**a**). The segments containing different sequences (Black, Red, Blue) are denoted with dashed rectangles. A schematic of the DNA substrate indicates the various DNA sequences and the relative positions of the extrahelical loop and the probe. **c** Left, a schematic of the assay. Right, polar plot of reads from GLOE-seq performed on pSam_(T)₄ with the extrahelical loop on the top strand reacted with the indicated proteins. The strand discontinuities in the looped strand are shown in red, and in the opposite strand in black. Source data are provided as a Source Data file.

loop, a G/T mismatch or relaxed DNA with or without a nick did not trigger the activity of MutSβ-MutLγ (Supplementary Fig. 5a–d).

### Loop-triggered DNA incision sites by MutSβ-MutLγ extend unidirectionally

The MutSβ-MutLγ-dependent incision sites on the DNA strand opposite the loop were preferentially located on one side of the loop. When we used the P2 probe complementary to the left side of the ScaI restriction site, we observed DNA fragments largely smaller than 1.4 knt, as indicated by the orange triangles (Fig. 1j). Reciprocally, when we used the P1 probe that anneals to the other side of the ScaI cleavage site, larger fragments were observed, again corresponding to incision sites on the left side opposite the loop (Fig. 1k). The experiments above suggested a model where MutSβ identifies the looped DNA structure and recruits MutLγ. MutLγ then slides away from the loop or forms a filament-like structure along DNA preferentially toward the 5' end on the non-looped strand[46], where it cuts DNA. Our data demonstrate that PCNA, which can get loaded at strand discontinuities but also at DNA loops[40], may act as an obstacle to block the sliding of MutLγ (Fig. 1g and Supplementary Figs. 4a, b, 5e). To substantiate this model, we placed catalytically-dead Cas9 (dCas9) as a protein block 171 bp upstream the loop and monitored DNA cleavage by MutSβ-MutLγ (Fig. 1l). Similarly to the reactions with PCNA, dCas9 also restricted the positions of the DNA incision sites past dCas9, likely because it inhibited the sliding of MutLγ (Fig. 1l, dashed rectangle in lane 4, Supplementary Fig. 5f). Instead, more frequent incisions were observed between the loop and the dCas9 block, where MutLγ is thought to accumulate (Fig. 1l, green triangle). Higher nucleosomal density within repeat-containing regions was found to reduce repeat expansion[47], suggesting that restricting the sliding of MutLγ along the DNA may reduce expansion rates. The preferential positions of the DNA incision sites 5' from the loop identified here would allow DNA

displacement synthesis to proceed in the 5'→3' direction toward the loop, as shown in the next section.

### MutSβ-MutLγ cut DNA with a moderate sequence preference

A notable observation from the experiments with MutSβ-MutLγ was that the DNA incision sites were non-randomly distributed, resulting in a distinct DNA cleavage pattern observed on Southern blots (Fig. 1g, j). We envisioned two scenarios. MutLγ may somehow measure the physical distance from the extrahelical loop, which could be related to the properties of its DNA sliding capacity or a propensity to form filaments of a finite length[46]. The second possibility was that MutLγ prefers to cleave certain DNA sequences. To distinguish between these two models, we replaced a segment of the DNA sequence on the left side of the loop with other unrelated DNA sequences of the same length (Fig. 2a, b). The overall efficacy of DNA cleavage moderately differed in each case (Fig. 2a), and the resulting DNA cleavage pattern corresponding to various DNA segments was distinct in each case, supporting the second scenario (Fig. 2a, b).

To define the positions of nicks at single-nucleotide resolution, we utilized GLOE-seq[48]. The method is based on the ligation of DNA adapters to the 3' DNA ends at nick sites, followed by amplification and next generation sequencing (Supplementary Fig. 6a). In accord with the low-resolution of Southern blotting, we observed non-random distribution of incision sites in the DNA strand opposite the loop (in black), while the looped strand was incised much less (in red) (Fig. 2c). The NGS analysis also better illustrated the non-random positions of the DNA cleavage sites, illustrated by distinct peaks, which preferentially extended from the loop in the 5' direction (Fig. 2c). The apparent sequence preference could explain why various alleles of the *Huntingtin* gene differing in the sequences adjacent to the CAG repeats undergo distinct expansion rates[49].

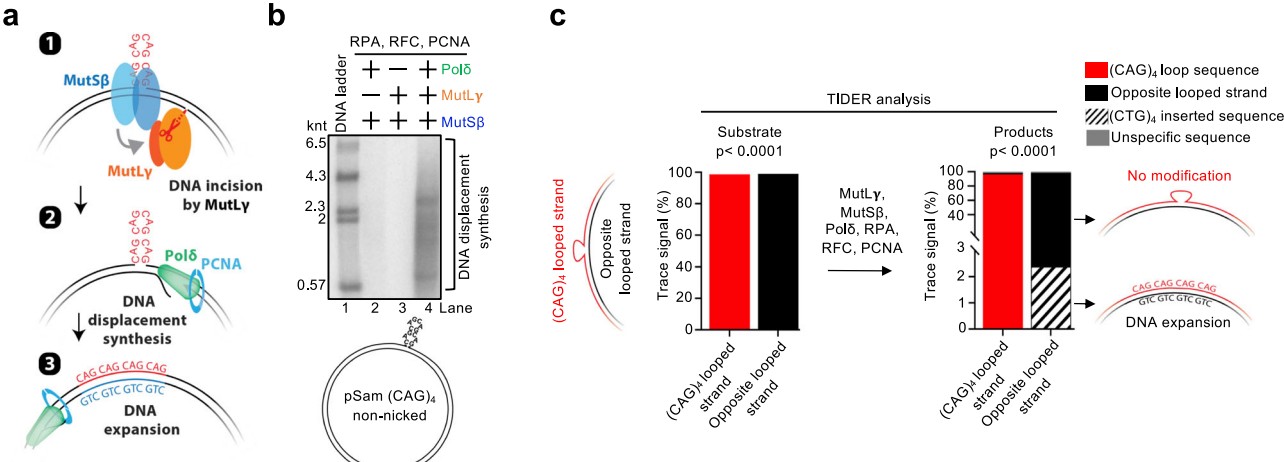

**Fig. 3 | MutSβ, MutLγ, RPA, RFC, PCNA and Polδ catalyze repeat expansion. a** A model for trinucleotide repeat expansion. **b** A representative of three independent DNA strand displacement assays. pSam_(CAG)₄ DNA was nicked by MutSβ and MutLγ, and the reaction was then supplemented with Polδ, RPA, RFC and PCNA. Strand displacement DNA synthesis was detected by the incorporation of radioactive dCTP into high molecular weight DNA. **c** pSam_(CAG)₄ DNA was reacted with the indicated proteins, followed by Sanger and TIDER analysis of the reaction products. While no modification of the looped strand was observed, a fraction of the opposite strand contained an insertion of the (CTG)₄ sequence, supporting the model depicted in (**a**). The *p*-value associated with the trace signal is computed using a two-tailed *t* test, with the standard errors derived from the variance-covariance matrix. Source data are provided as a Source Data file.

## MutSβ, MutLγ, RPA, RFC, PCNA and Polδ catalyze repeat expansion

Polδ was found to significantly promote repeat expansion, implicating DNA synthesis in this process[7]. The position of the MutLγ DNA incision sites with respect to the extrahelical loop observed in our in vitro assays could allow DNA displacement synthesis to proceed 5′→3′ toward the loop. DNA synthesis with the loop in the top template DNA strand would lead to DNA expansion of the bottom strand (Fig. 3a). In accord with this model, we observed that DNA incision sites produced by MutSβ-MutLγ indeed represent suitable entry points for DNA displacement synthesis by Polδ in conjunction with RPA, RFC and PCNA, as scored by the incorporation of radioactive dCTP into high molecular weight DNA (Fig. 3b). We next performed Sanger sequencing combined with TIDER analysis to identify the nature of the expanded sequence[50]. We detected no modifications in the top strand containing the loop (Fig. 3c). In contrast, the trace signal from the strand opposite the loop exhibited a statistically significant 12 nt-long inserted sequence identified as (CTG)₄, which corresponds to the site of the extrahelical loop (Fig. 3c). Importantly, Polδ-mediated DNA strand displacement synthesis leaves the displaced flap intact, which could inhibit further progression of strand displacement and/or reduce the proportion of the nascent strand detected during sequencing. By supplementing the reaction with the flap-removing enzyme DNA2, the (CTG)₄ insertion frequency increased to 15.5% (Supplementary Fig. 6b). Our experiments demonstrate that MutSβ, MutLγ, RPA, RFC, PCNA, and Polδ constitute the minimal enzymatic system capable to catalyze trinucleotide repeat expansion in vitro.

## RFC and PCNA promote FAN1 to cleave looped DNA strands

A key factor that protects from triplet repeat expansion is FAN1[6,7]. Its role in preventing triplet expansion is dependent on its nuclease activity[17,33–36,51,52]. FAN1 specifically interacts with MLH1[26,32,53], and this interaction was found to be important for preventing triplet expansion[32,53]. The FAN1 nuclease activity was observed to be activated by PCNA loaded at nicks on a looped DNA substrate, resulting in additional cuts to the nicked strand[37]. Using a covalently closed plasmid with a (CAG)₄ extrahelical loop, we observed that FAN1 on its own degraded DNA, due to the combination of its endonuclease and exonuclease activities, leading to the decrease of substrate scDNA (Supplementary Fig. 7a, b, lane 2). The observed DNA cleavage was

dependent on the nuclease activity of FAN1, as reactions with a FAN1 variant bearing three substitution mutations in the nuclease active site did not show any activity (FAN1 ND, D960A, D981A, R982A)[27,54] (Supplementary Fig. 7a and 7b, lane 6). In the presence of RFC and PCNA, the activity of FAN1 was enhanced as estimated by further disappearance of the substrate, and almost only nicked DNA was observed (Supplementary Fig. 7b, lane 5), suggesting that RFC and PCNA stimulate the endonuclease activity and restrict the exonuclease activity of FAN1. As with MutSβ-MutLγ, a fraction of the DNA substrate was linearized by FAN1 (Supplementary Fig. 7b, c). Notably, the regulation of FAN1 was already observed in the presence of RFC alone (Supplementary Fig. 7b, lane 3). Therefore, in contrast to PCNA-regulated canonical mismatch repair reactions, where RFC functions only as a PCNA loader[9,42], RFC directly controls FAN1. To test whether FAN1 directly interacts with RFC, we performed a FAN1 pull-down assay (Fig. 4a). MBP-tagged FAN1 was immobilized on amylose resin and incubated with RFC alone, PCNA alone, or both proteins together. In addition to the previously reported interaction between FAN1 and PCNA[37], we now demonstrated that RFC also binds directly to FAN1 (Fig. 4a). The direct interaction of RFC and PCNA was also predicted based on AlphaFold-Multimer-based modeling[55]. The inhibition of the exonuclease activity of FAN1 by RFC is reminiscent of RFC-dependent inhibition of the exonuclease activity of EXO1 starting at a nick[56]. Thus, RFC's regulation of FAN1 likely involves both direct protein-protein interactions and DNA binding. PCNA further stimulated FAN1 (Supplementary Fig. 7b, compare lanes 3 and 5), but it had no role on its own without RFC (Supplementary Fig. 7b, lane 4)[37]. The stimulation by PCNA involved ATP hydrolysis (Supplementary Fig. 7b, c, compare lanes 3 and 5). The regulation of FAN1 by RFC and PCNA required species-specific interactions, as budding yeast RFC or PCNA could not substitute the cognate human proteins (Supplementary Fig. 7d). We confirmed that the yeast homolog proteins are active under our experimental conditions at 37 °C, as demonstrated by Polδ-dependent DNA strand-displacement synthesis, an activity known to be stimulated by RFC-PCNA and Pif1 (Supplementary Fig. 7e, f)[57,58]. We note that in these reactions and in those presented below (Fig. 4b–e, Supplementary Fig. 7b–h), RFC and PCNA strongly promoted FAN1 activity on looped DNA strands without a need for a strand discontinuity, as PCNA can get loaded on looped DNA[40,42].

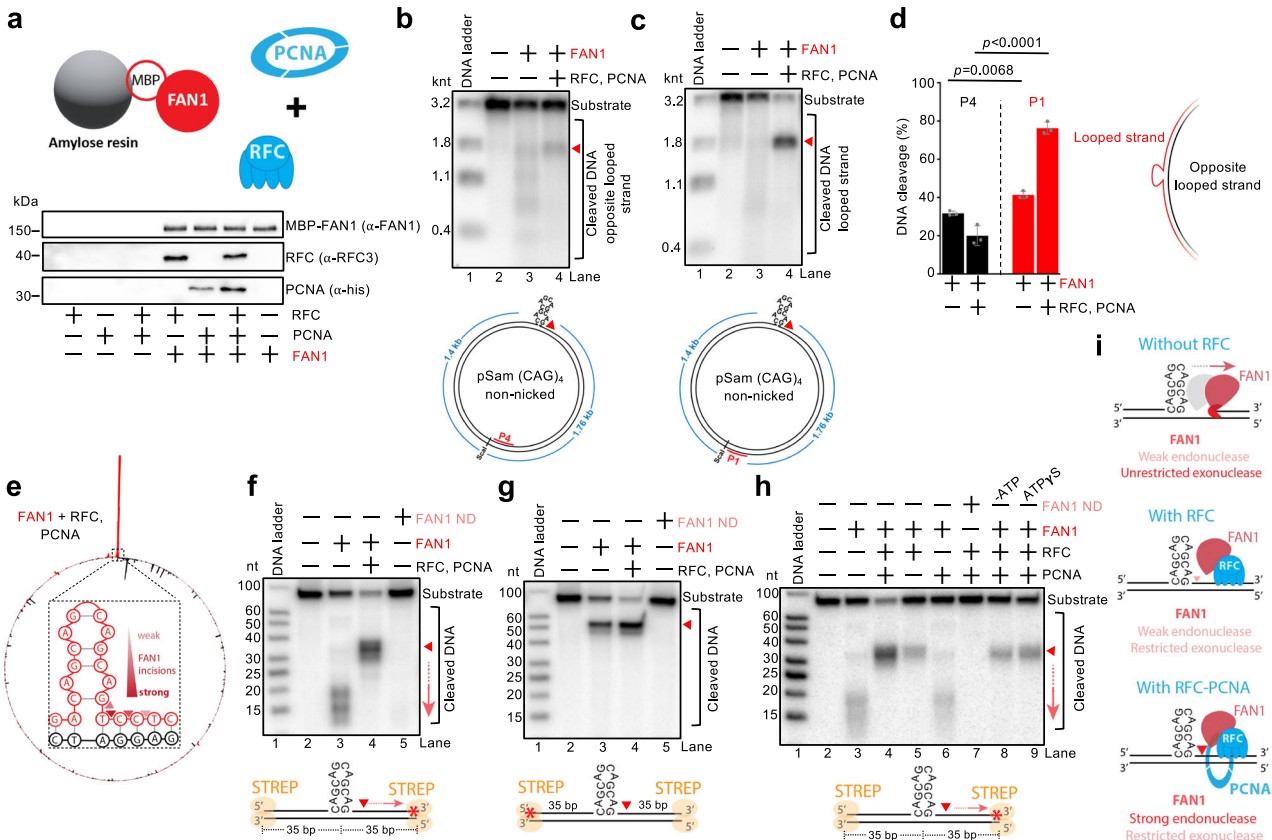

**Fig. 4 | RFC and PCNA regulate FAN1 DNA cleavage at 3' ends of extrahelical loops. a** Top, a cartoon depicting the pulldown assay. Bottom, a representative of three pulldown experiments where FAN1 was immobilized on amylose resin using MBP-tagged FAN1, and incubated with either RFC, PCNA or both. **b, c** Representative nuclease assays with pSam_(CAG)$_4$ DNA and the indicated proteins. The reaction products were analyzed by Southern blotting with probes complementary to the strand opposite the loop, P4 (**b**) or complementary to the looped strand, P1 (**c**). The red triangle indicates cleavage position at the loop site. Source data are provided as a Source Data file. **d** Quantification of Southern blot-based nuclease assays, such as shown in (**b, c**). Statistical significance was determined by two-way ANOVA followed by Tukey's multiple-comparisons test comparing FAN1 cleavage on the strand opposite the loop versus the strand that contains the loop in the presence or absence of RFC and PCNA. Averages shown; error bars, s.e.m.; $n = 3$ independent experiments. With RFC and PCNA, FAN1 preferentially cleaves the looped DNA strand. **e** Polar plot of reads obtained with GLOE-seq on pSam_(CAG)$_4$ reacted with the indicated proteins. In the center of the plot, a

magnified view indicates the positions of the main DNA incisions by FAN1 with RFC and PCNA. Compare with Supplementary Fig. 7i. **f, g** Oligonucleotide-based nuclease assays with the indicated proteins. Nuclease-deficient FAN1 ND contains D960A, D981A, R982A substitutions. Reaction products were analyzed by denaturing polyacrylamide gels. Top, representative assays. Bottom, a cartoon of the respective DNA substrates. The DNA ends were blocked by monovalent streptavidin (STREP). The red asterisk (*) represents the position of the radioactive label. In (**f**), the looped DNA strand was 3'-labeled, in (**g**), the looped DNA strand was 5'-labeled. The red triangle indicates the position of the endonucleolytic DNA incision by FAN1. The red arrow indicates subsequent exonucleolytic DNA degradation by FAN1 that is restricted by RFC and PCNA. **h** Assays as in (**f, g**), with the indicated proteins showing the individual effects of RFC and PCNA on FAN1. Lane 8, ATP was omitted. Lane 9, non-hydrolysable ATPγS was used instead of ATP. **i** A cartoon showing that RFC primarily restricts the exonuclease activity of FAN1, while PCNA, loaded on DNA by RFC, stimulates the endonuclease activity of FAN1. Source data are provided as a Source Data file.

To distinguish between the looped and non-looped DNA strands, we next carried out Southern blotting with strand-specific probes. We observed that RFC and PCNA stimulated FAN1 to specifically cleave the looped DNA strand, while only weak cleavage was detected on the opposite strand (Fig. 4b–d). As above, moderate stimulation of incision of the looped strand was observed when FAN1 was combined with RFC alone, and PCNA then strongly enhanced this effect (Supplementary Fig. 7g, h). The results were confirmed by GLOE-seq of the reaction products, indicating that the primary incision by FAN1 in the presence of RFC-PCNA is located one nucleotide downstream of the extrahelical loop, while much less frequent cuts were observed elsewhere on the DNA (Fig. 4e). Without RFC-PCNA, the incision past the loop was less prominent compared to other incision sites on either the looped or the opposite looped DNA strand (Supplementary Fig. 7i). Importantly, the preferred incision of the looped DNA by FAN1-RFC-PCNA occurred on a DNA substrate without a strand discontinuity (Fig. 4b and Supplementary Fig. 7h), in accord with FAN1 acting on DNA structures in quiescent cells, where strand discontinuities due to a

lack of replication are thought to be rare. In order to further illustrate the different DNA cleavage of (CAG)$_4$-looped substrate by MutSβ-MutLγ versus FAN1, we performed GLOE-seq with MutSβ-MutLγ both in the presence and absence of RFC-PCNA (Supplementary Fig. 7j). The cleavage pattern of MutLγ on (CAG)$_4$ closely mirrored that observed on the (T)$_4$ plasmid substrate (compare Fig. 2c and Supplementary Fig. 7j, left), showing a fundamental difference compared to FAN1, both in the position of the DNA incision sites, as well as with respect to RFC-PCNA, which inhibit MutSβ-MutLγ and stimulate FAN1 (Fig. 4e and Supplementary Fig. 7i, j).

To further complement the data, we next utilized oligonucleotide-based DNA substrates containing a 12-nucleotide (CAG)$_4$ extrahelical loop in the top strand, with radioactive labels at various positions. Streptavidin was placed at both DNA ends to inhibit the exonuclease activity of FAN1. FAN1 incised the looped DNA strand immediately downstream of the loop (Fig. 4f, g and Supplementary Fig. 8a, b). In the absence of RFC and PCNA, this initial incision was followed by 5' → 3' exonucleolytic degradation (Fig. 4f). However, when both RFC and

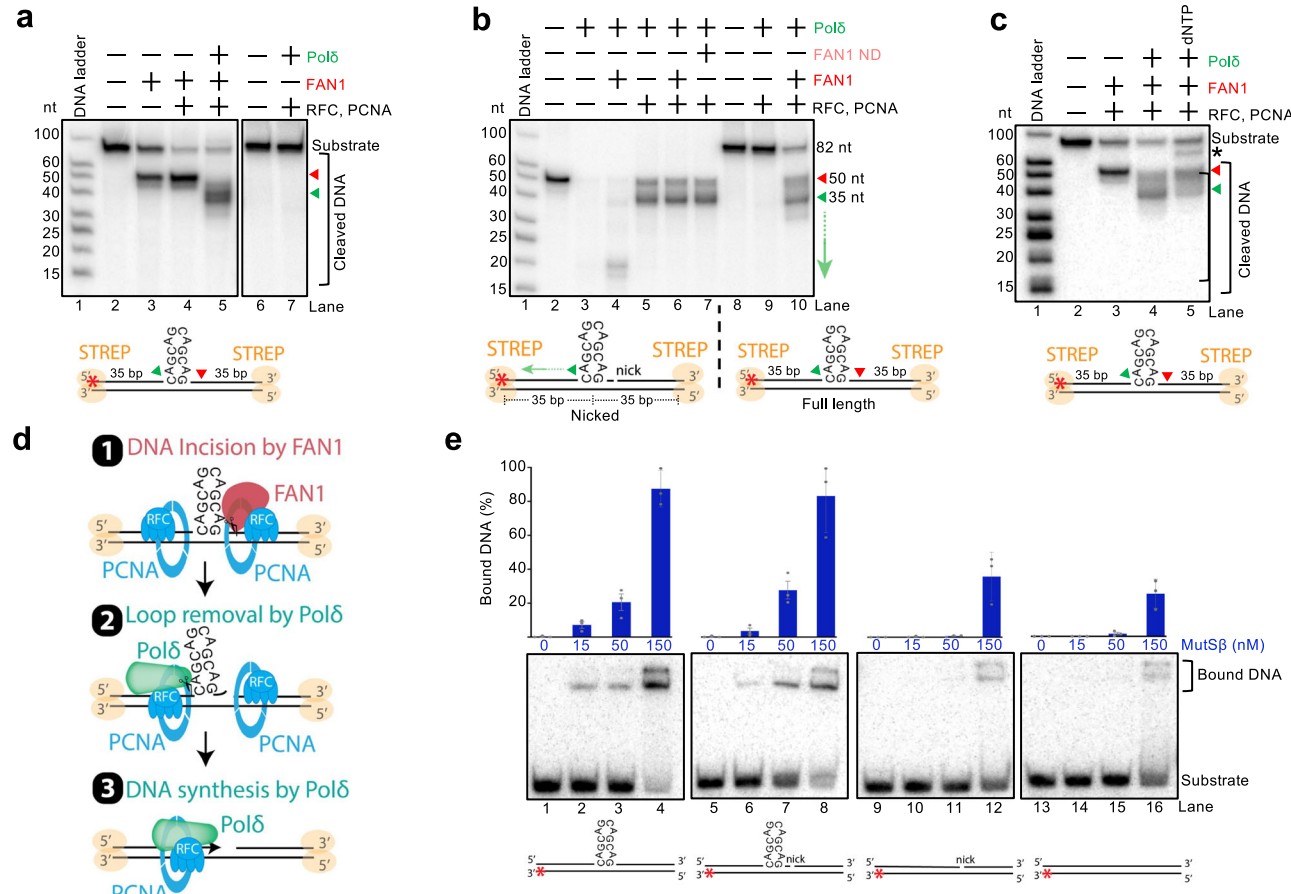

**Fig. 5 | RFC, PCNA, FAN1 and Polδ excise extrahelical DNA loops. a–c** Top, a representative of three independent oligonucleotide-based nuclease assays with the indicated proteins analyzed by denaturing polyacrylamide gels. Bottom, a cartoon of the used DNA substrates. The looped DNA strand was 5'-labeled, as indicated by the red asterisk (*). The DNA ends were blocked by monovalent streptavidin (STREP). Panel (**a**) shows the position of the initial DNA incision by FAN1-RFC-PCNA, indicated by the red triangle, which is followed by the degradation of the looped DNA by Polδ-RFC-PCNA up to the point indicated by the green triangle. Panel (**b**) shows that FAN1 activity can be bypassed by placing a strand discontinuity downstream of the loop, and that Polδ's exonuclease activity is restricted by RFC and PCNA to the DNA loop. Nuclease-deficient FAN1 ND contains D960A, D981A, R982A substitutions. Panel (**c**) shows that DNA incisions by FAN1-RFC-PCNA (red triangle) and Polδ-RFC-PCNA (green triangle), occur in the presence of dNTPs followed by DNA synthesis (black asterisk), leading to the synthesis of the top DNA strand without a loop. **d** A cartoon depicting how FAN1 and Polδ, assisted by RFC and PCNA, excise DNA loops. **e** Representative electrophoretic mobility shift assay performed with four different substrates, from left to right: (CAG)₄ loop, (CAG)₄ loop with a nick, nicked double strand DNA or double strand DNA, with MutSβ at the indicated concentrations. Averages shown; error bars, s.e.m.; *n* = 3 independent experiments. For each substrate, the bottom strand was 3'-labeled, as indicated by the red asterisk (*). Source data are provided as a Source Data file.

PCNA were present, the exonuclease activity of FAN1 was suppressed, while the endonuclease activity of FAN1 was enhanced (Fig. 4f, compare lanes 3 and 4). RFC alone was capable of inhibiting FAN1's exonuclease activity (Fig. 4h, compare lanes 3 and 5), whereas PCNA by itself had no notable effect (Fig. 4h, compare lanes 3 and 6). Thus, PCNA must likely be loaded onto DNA by RFC to stimulate FAN1's endonuclease activity. Supporting this notion, no stimulation was observed when ATP was absent or when a nonhydrolyzable ATP analog was used (Fig. 4h, lanes 8 and 9), as PCNA loading by RFC requires ATP hydrolysis. Together, these results show that both RFC and PCNA are involved in restraining FAN1's exonuclease and enhancing its endonuclease activity (Fig. 4i). We note the regulation of FAN1's activity likely involves a specific interplay with both human RFC and human PCNA, as neither factor could be replaced with their budding yeast *S. cerevisiae* homologs (Supplementary Fig. 8c, d). In summary, on (CAG)₄ DNA substrates, the FAN1 nuclease in conjunction with RFC and PCNA, cleaves the DNA strand opposite the one targeted by MutSβ-MutLγ (compare Figs. 2c and 4e). This difference likely reflects the contrasting roles of these factors in triplet repeat metabolism[7].

## FAN1·RFC·PCNA with Polδ catalyze repeat contraction

Considering the involvement of DNA synthesis in triplet repeat metabolism, we next included Polδ in our reconstituted reactions with the oligonucleotide-based looped DNA substrate. Polδ, together with FAN1-RFC-PCNA, catalyzed the formation of a product of less than 40 nucleotides when 5'-labeled substrate was used, suggesting that the 3'→5' exonuclease activity of Polδ efficiently excised the loop while the rest of the top strand remained intact (Fig. 5a, compare lanes 4 and 5). Polδ with RFC-PCNA had no activity on its own, showing that it acts downstream of FAN1 (Fig. 5a, lanes 6 and 7). We next used a substrate with a pre-existing nick 3' of the loop to bypass the activity of FAN1 (Fig. 5b). We observed that Polδ could still remove the loop in the absence of FAN1 (Fig. 5b, compare lanes 2 and 5). Without RFC-PCNA, Polδ continued degrading the top strand, while with RFC-PCNA, Polδ's 3'→5' exonuclease was restricted once the loop sequence was excised (Fig. 5b, compare lanes 3 and 5). The reaction end products on pre-nicked DNA with Polδ and RFC-PCNA mirrored those on substrates without the nick, but when FAN1 was additionally present (Fig. 5b, compare lanes 5 and 10). Similar data were obtained when the substrate was first processed by FAN1-RFC-PCNA, the proteins were heat-

denatured, and Polδ was added. Also in this case, uncontrolled Polδ's 3'→5' exonuclease degraded the top strand beyond the loop (Supplementary Fig. 8e). The restriction of the human Polδ's 3'→5' exonuclease activity required specific interplay with human RFC and human PCNA, because the human factors could not be replaced by yeast Polδ, yeast RFC or yeast PCNA (Supplementary Fig. 8e, f). Thus, by specifically moderating both FAN1's 5'→3' and Polδ's 3'→5' exonuclease activities, RFC and PCNA ensure that the excision remains tightly confined to the loop (Supplementary Fig. 8g).

The capacity of Polδ-RFC-PCNA to excise the loop downstream of FAN1-RFC-PCNA was largely unaffected by the presence of dNTPs (Fig. 5c, lane 5). When dNTPs were included and DNA synthesis was allowed, we observed a $^{32}$P-labeled product corresponding to the size of the full-length oligonucleotide with the loop sequence removed (Fig. 5c, lane 5, black asterisk).

In summary, FAN1, RFC, PCNA, and Polδ constitute the minimal enzymatic system capable of loop excision in vitro. The reaction occurs in three steps (Fig. 5d). First, FAN1-RFC-PCNA cleaves the DNA strand 3' of the loop. RFC-PCNA stimulates the efficacy of the endonucleolytic DNA incision by FAN1 and simultaneously restricts its 5'→3' exonuclease activity. Second, Polδ's 3'→5' exonuclease activity removes the loop, and RFC-PCNA controls that the looped strand is subsequently not over-resected. Third, Polδ's polymerase activity resynthesizes the DNA strand without the loop sequence, in a reaction stimulated by RFC and PCNA. Our assays identify components and provide a likely mechanistic basis for triplet repeat contraction observed in cell extracts and cellular experiments[37,59,60].

### FAN1 inhibits MutSβ-MutLγ through structural and catalytic functions

Cellular experiments revealed that the protective function of FAN1 against triplet repeat expansion is dependent on a combination of its nuclease function and its capacity to interact with MLH1[17,32–36,51,61], although mechanistic insights were lacking. We showed that the endonucleolytic cut by FAN1 just downstream of the loop primes the removal of the extrahelical loop by Polδ. The model predicts that loop removal by FAN1-Polδ-RFC-PCNA would prevent recognition by MutSβ, which in turn fails to activate MutLγ. To determine whether DNA after loop excision remains a substrate for MutSβ, we performed electrophoretic mobility shift assays. Whereas MutSβ bound looped DNA and DNA with a nick at the 3' end of the loop to mimic the result of FAN1 incision (Fig. 5e, lanes 1–8), MutSβ bound much less efficiently to DNA where the loop was absent, such as nicked or fully duplex DNA (Fig. 5e, lanes 9–16). These results are consistent with the models where loop removal by FAN1's nuclease activity in conjunction with Polδ-RFC-PCNA prevents the activation of MutSβ-MutLγ (Supplementary Fig. 5c, d).

To explore a potential structural role of FAN1 in antagonizing MutLγ, we constructed a mutant by disrupting two adjacent motifs in FAN1 known to interact with MLH1, MIP and MIM[32,53], both in nuclease proficient and nuclease-deficient FAN1 backgrounds (Supplementary Fig. 8h). We then performed nicking assays with these proteins on a covalently closed plasmid with a (CAG)$_4$ extrahelical loop. We observed that nuclease-deficient FAN1 inhibited the nuclease activity of MutSβ-MutLγ, while nuclease-deficient FAN1 MIP-MIM was not inhibitory (Fig. 6a). In pulldown experiments, we confirmed that FAN1 MIP-MIM fails to physically interact with MLH1[32] (Supplementary Fig. 8i). Moreover, we observed that the presence of FAN1 reduced the interaction between recombinant MutSβ and MutLγ (Fig. 6b). Such disruption was not seen when FAN1 MIP-MIM was used (Fig. 6b)[53]. The direct interaction between FAN1 and MLH1 thus restricts the MutLγ nuclease, likely by preventing its activation by MutSβ (Fig. 6a, b)[53].

We next turned to Southern blots with strand-specific probes. Looking at the bottom, non-looped DNA strand, we observed that the pathological incision by MutSβ-MutLγ was suppressed by nuclease-deficient or also nuclease-proficient FAN1 (Fig. 6c, orange triangles). On the looped strand, we found that conversely the nuclease activity of FAN1 was not restricted by MutSβ-MutLγ (Fig. 6d). The experiments also showed that the nuclease activities of FAN1 and FAN1 MIP-MIM per se were indistinguishable (Fig. 6d). Our reconstituted experiments thus provide a mechanistic basis for both catalytic and structural roles of FAN1 in preventing pathological DNA incisions by MutLγ (Supplementary Fig. 8j).

Finally, we reacted FAN1, MutSβ, MutLγ, RFC, PCNA, RPA and Polδ with the (CAG)$_4$ extrahelical loop plasmid and performed Sanger sequencing of both the top (looped) and bottom (non-looped) DNA strands. Strikingly, more than 70% of the trace signal from the looped strand exhibited DNA shortening, with the vast majority of DNA molecules losing the (CAG)$_4$ loop-out (Fig. 6e). In contrast, no modifications of the bottom DNA strand were detected (Fig. 6e). These results support our model where FAN1 together with Polδ's exonuclease activity, guided by PCNA and RFC, remove the extrahelical loop, and Polδ synthesizes the DNA strand without the loop sequence, leading to repeat contraction. Simultaneously, FAN1 directly inhibits MutSβ-MutLγ activity. In summary, we showed that FAN1 prevents pathological MutSβ-MutLγ-dependent cutting of the DNA strand opposite the loop by a combination of its structural (FAN1-MLH1 interaction-dependent) and nucleolytic functions. In addition, FAN1, in conjunction with Polδ and PCNA, can excise DNA loops, leading to repeat contraction (Supplementary Fig. 8j).

## Discussion
Triplet repeat instability underlies numerous human disorders and can arise through diverse pathways, including those active in non-dividing brain cells. Repetitive DNA regions are inherently dynamic and prone to forming secondary structures even independently of DNA replication. Among these, staggered loop-outs (S-form structures) prevalent in long triplet repeats may represent key physiological substrates for the nuclease activities of MutLγ and FAN1[62]. Using reconstituted biochemical reactions, we demonstrate that MutSβ stimulates MutLγ to introduce nicks in looped DNA substrates in a highly biased manner. First, as shown by Kadyrov and colleagues[25], MutLγ primarily cuts the DNA strand opposite extrahelical loops. We also show that the nicks are preferentially located 5' from the loop-out, resulting from likely directional sliding of MutLγ activated by MutSβ away from the loop-out. PCNA limits the sliding of MutLγ and restricts the incision positions to the vicinity of the loop. The preferential 5' orientation of the nicks relative to the loop allows Polδ, assisted by RFC, PCNA and RPA, to displace and synthesize DNA toward the loop. Because the looped strand is used as a template, the reaction results in DNA extension[7]. We suggest that this process can occur outside of S-phase, promoting triplet repeat expansions even in the absence of conventional DNA replication in non-dividing cells (Fig. 7a)[7]. In proliferating cells, such nicks might cause replication fork collapse leading to one-ended breaks, ultimately resulting in sudden repeat length changes e.g., via break-induced replication (Fig. 7b)[63]. We demonstrate that MutSβ, MutLγ, RFC, PCNA, and Polδ are the minimal components required to reconstitute DNA expansion in vitro.

In contrast to MutLγ, that drives DNA expansion, FAN1 is protective. FAN1's anti-expansion activity depends on its nuclease function, but also its ability to interact with the MLH1 subunit of MutLγ[32,53]. Previously, it was demonstrated that PCNA, loaded by RFC, stimulated DNA incisions of looped DNA substrates with strand discontinuities[37]. We show that PCNA drives FAN1 incisions to a site precisely at the 3' end of the loop. Importantly, the reaction does not require a pre-existing nick, as PCNA can get loaded by RFC at sites of the DNA loop-outs. Moreover, we show that while PCNA activates the endonuclease of FAN1, RFC restricts its exonuclease activity, in a highly species-specific manner, revealing novel functional interactions. Previously, a loop excision activity dependent on FAN1 and additional unknown

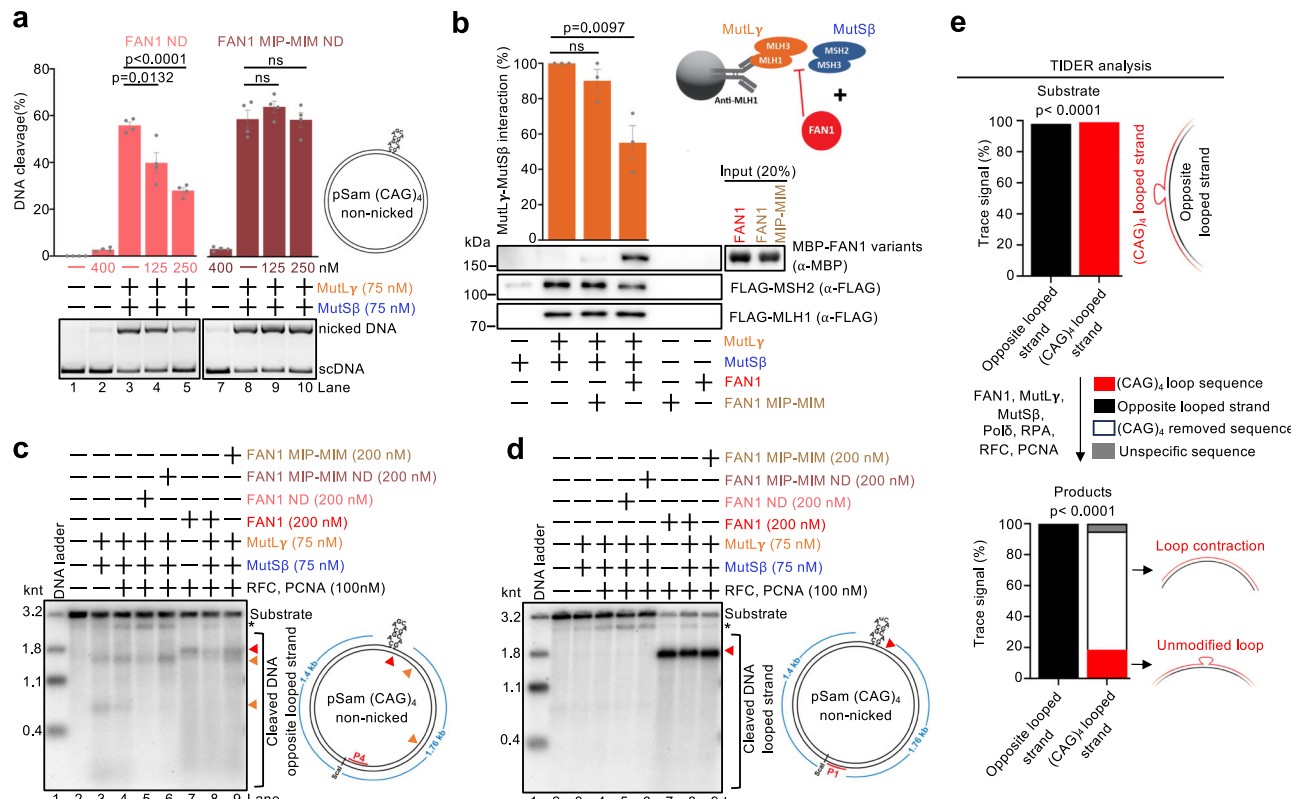

**Fig. 6 | FAN1 inhibits MutSβ-MutLγ through structural and catalytic functions.**
**a** Representative nicking assays with pSam_(CAG)₄ DNA and the indicated proteins. Averages shown; error bars, s.e.m.; $n = 4$ independent experiments. Statistical analysis was performed with a two-tailed *t* test. Nuclease-dead FAN1 ND contains D960A, D981A, R982A substitutions. Nuclease-dead and MLH1-binding defective FAN1 MIP-MIM ND contains Y128A, F129A, L155A, L159A, D906A, D981A, R982A point mutations. FAN1 ND inhibits DNA cleavage by MutSβ-MutLγ. **b** Left, protein interaction assay. MutLγ (MLH1-MLH3) was immobilized using anti-MLH1 antibody, followed by incubation with MutSβ (MSH2-MSH3) and/or FAN1 variants, as indicated. MLH1-binding defective FAN1 MIP-MIM contains Y128A, F129A, L155A, L159A point mutations. When FAN1 was included (note that MBP-tagged FAN1 was used), the interaction between MutLγ and MutSβ was diminished. There was no effect on MutLγ and MutSβ interaction when MBP-tagged FAN1 MIP-MIM was used. Top, averages shown; error bars, s.e.m.; $n = 3$ independent experiments. The data were normalized to the interaction between MutLγ and MutSβ without FAN1. Statistical analysis was performed with a two-tailed *t* test. ns, non-significant. Bottom,

representative Western blot analysis. The cartoon on the right depicts FAN1 disrupting the physical interaction between MutLγ and MutSβ. **c**, **d** A representative of three independent nuclease assays with pSam_(CAG)₄ DNA and the indicated proteins. The reaction products were analyzed by Southern blotting with probes complementary to the strand opposite the loop, P4 (**c**) or to the looped strand, P1 (**d**). FAN1 MIP-MIM, MLH1-binding defective FAN1. FAN1 MIP-MIM ND, nuclease-dead and MLH1-binding defective FAN1. FAN1 ND, nuclease-dead. On the strand opposite the loop (**c**), the major DNA incision points by MutLγ are indicated with the orange triangles and an incision point by FAN1 is indicated by the red triangle. DNA incision by MutLγ is inhibited by FAN1 ND (lane 5). The looped strand (**d**) is cleaved efficiently by FAN1 (lanes 7–9), and its activity is not inhibited by MutLγ. *, DNA not cleaved by ScaI. **e** pSam_(CAG)₄ DNA was reacted with the indicated proteins in the presence of dNTPs, and the reaction products were analyzed by Sanger sequencing and TIDER analysis. The *p*-value associated with the trace signal is computed using a two-tailed *t* test, with the standard errors derived from the variance-covariance matrix. Source data are provided as a Source Data file.

factors was observed in cell extracts[37]. We identified that the initial DNA incision by FAN1 creates a substrate for Polδ's exonuclease activity, which removes the loop. Hence, FAN1, Polδ, RFC and PCNA are the minimal components required for loop excision in vitro. By eliminating DNA loop-outs, FAN1 prevents loop recognition by MutSβ and subsequent MutLγ activation. Furthermore, we observed that FAN1 prevented MutLγ nuclease activation by MutSβ, in accord with its ability to disrupt the physical interaction between MutSβ and MutLγ[53], explaining the additional structural component of FAN1 in the protection against DNA expansion. Our reconstituted reactions illuminate mechanisms underlying the instability of trinucleotide repeats and reveal biochemical reactions amenable to therapeutic interventions.

## Methods

### Cloning, expression and purification of recombinant proteins
The human MSH2-MSH3 (MutSβ), MSH2-MSH6 (MutSα), MSH4-MSH5 (MutSγ), MLH1-PMS2 (MutLα), MLH1-MLH3 (MutLγ) and MLH1-MLH3 3ND (MLH3 D1223N, Q1224K and E1229K, nuclease-deficient) heterodimers were expressed in *Spodoptera frugiperda* 9 (*Sf*9) insect cells using the Bac-to-bac expression system (Invitrogen) and the following

vectors, respectively: pFB-FLAG-PP-hMSH2, pFB-His-hMSH3, pFB-His-hMSH6, pFBDM-hMSH4co-STREP-hMSH5co-His, pFB-FLAG-hMLH1co, pFB-His-PMS2, pFB-MBP-PP-hMLH3co and pFB-MBP-PP-MLH3co3ND[23]. The *MLH1, MLH3, MSH4* and *MSH5* sequences were codon-optimized for expression in insect cells[23]. The human heterodimers were subsequently purified through affinity chromatography, using the N-terminal FLAG-tag on MSH2 or MLH1, the C-terminal STREP-tag on MSH4, the N-terminal 6xHis-tag on MSH3, MSH6 or PMS2, the C-terminal 8 x His-tag on MSH5 and the N-terminal MBP-tag on MLH3[23,64].

Human PCNA was expressed in *E. coli* cells from the pET23C-his-hPCNA vector[23]. Human RFC was expressed in *Sf*9 cells using the pFBDM-MBP-RFC1-RFC2-3-4-His-5 vector (Josef Jiricny, ETH Zürich)[23]. Human nuclease-deficient EXO1 (D173A) was expressed in *Sf*9 cells from the pFB-hEXO1(D173A)-FLAG construct[23]. Human RPA was expressed in *Sf*9 cells from the following vectors: pFB-RPA1, pFB-RPA2, and pFB-6xHis-RPA3. The purification was carried out by affinity chromatography, exploiting the N-terminal 6 x His-tag on the RPA3 subunit, followed by purification on HiTrap Blue HP column, HiTrap desalting column, and HiTrap Heparin columns (Cytiva)[65].

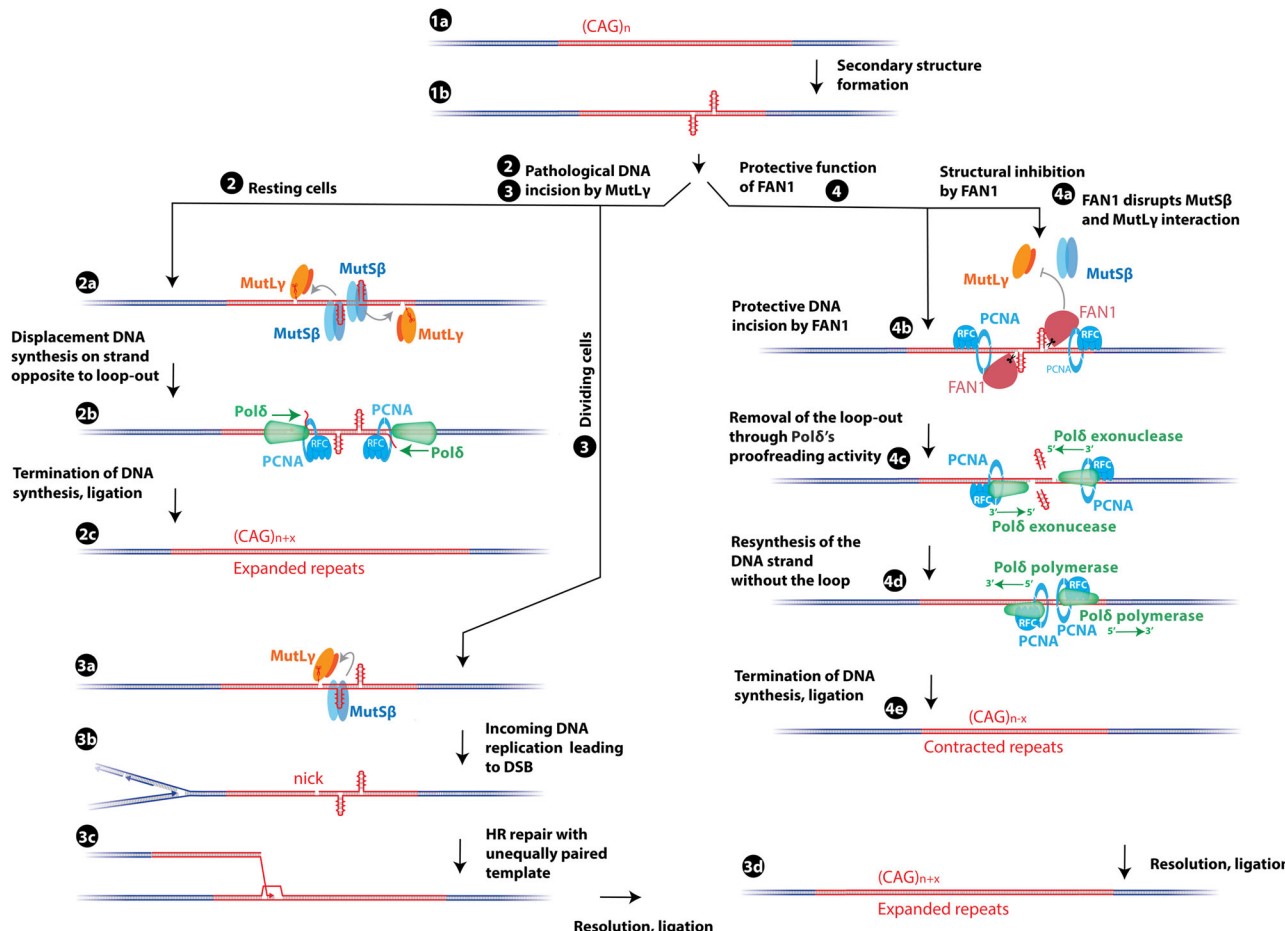

**Fig. 7 | Model for repeat expansion and contraction.** Models for DNA trinucleotide repeat expansion and contraction. DNA with (CAG)$_n$ repeats is prone to the formation of secondary DNA structures (1a, b). MutLγ promotes repeat expansion both in resting cells (2) and in dividing cells (3). Specifically, MutLγ introduces biased incisions opposite the loop-out (2a and 3a), primarily oriented in the 5′ direction relative to the loop-out. In resting cells, Polδ subsequently performs displacement DNA synthesis toward the loop-out, using it as a template (2b), thus resulting in repeat expansion (2c). In proliferating cells, DNA nicks produced by MutLγ within the trinucleotide repeats (3a) can trigger replication fork collapse, resulting in one-ended DNA breaks (3b). The consequent break-induced replication and unequal sister chromatid exchange (3c) can cause abrupt changes in repeat length (3d). FAN1 has a protective function against repeat expansion (4). On one hand, FAN1 inhibits the MutSβ-MutLγ nuclease activity by disrupting the MutSβ and MutLγ complex (4a), thereby structurally preventing pathological cleavage. In addition, in conjunction with its co-factors RFC and PCNA, FAN1 nuclease incises the loop-out DNA at the 3′ end of the loop (4b). The loop is subsequently removed by Polδ's proofreading activity (4c). Polδ then resynthesizes the strand without the loop (4d), resulting in contraction (4e).

Wild type human FAN1, nuclease-dead FAN1 ND (D960A, D981A, R982A)[27,54], MLH1-binding defective FAN1 MIP-MIM (Y128A, F129A, L155A, L159A)[32], and MLH1-binding defective and nuclease-dead FAN1 MIP-MIM ND (Y128A, F129A, L155A, L159A, D906A, D981A, R982A) were expressed in *Sf*9 cells from the respective pFastBac1 constructs. The pFB-MBP-FAN1-wt and pFB-MBP-FAN1 D906A plasmids were a kind gift from Josef Jiricny, ETH Zürich. The FAN1 variants used in of this study were generated by mutating the relevant residues by QuikChange II site-directed mutagenesis kit (Agilent Technologies) using the primers listed in Supplementary Table 1. pFB-MBP-FAN1 D906A was mutagenized using the oligonucleotides FAN1 FW D981A/R982A and FAN1 REV D981A/R982A, in order to generate pFB-MBP-FAN1-ND. pFB-MBP-FAN1-MIP-MIM was generated by two consecutive rounds of site-directed mutagenesis of pFB-MBP-FAN1-wt using the following oligos: FAN1 MIP FW, FAN1 MIP REV, FAN1 MIM FW, FAN1 MIM REV. pFB-MBP-FAN1-MIP-MIM-ND was generated by mutagenesis of pFB-MBP-FAN1-ND as described for pFB-MBP-FAN1-wt. Wild-type human FAN1 and its variants were purified following the same procedure. *Sf*9 cells were seeded at 5,000,000 cells/ml in 500 ml 18 h before infection. The cells were then infected with corresponding baculoviruses and incubated for 52 h at 27 °C with constant agitation. The cells were harvested by centrifugation (500 × g, 10 min) and washed once with PBS (137 mM NaCl, 2.7 mM KCl, 10 mM Na$_2$HPO$_4$, 1.8 mM KH$_2$PO$_4$). The pellets were snap-frozen in liquid nitrogen and stored at − 80 °C. All subsequent steps were carried out at 4 °C. Cell pellets were resuspended in 3 volumes of lysis buffer containing 50 mM Tris-HCl, pH 7.5, 5 mM β-mercaptoethanol (β-ME), 1 mM ethylenediaminetetraacetic acid (EDTA), 1 mM phenylmethylsulfonyl fluoride (PMSF), 1:300 (volume/volume) protease inhibitor cocktail (Sigma), 30 μg/ml leupeptin (Merck) and incubated for 20 min while stirring. Glycerol was added to a final concentration of 10%, NaCl was added to the final concentration of 305 mM. The suspension was further incubated for 30 min and centrifuged for 30 min at 55,000 × g to obtain the soluble extract. The supernatant was transferred to tubes containing pre-equilibrated amylose resin (New England Biolabs) and incubated for 1 h with continuous agitation. The resin was collected by spinning at 2000 × g for 2 min and washed 3 times batchwise with amylose wash buffer (50 mM Tris-HCl, pH 7.5, 2 mM β-ME, 1 mM PMSF, 10% glycerol, 1 M NaCl). The resin was transferred onto a disposable column (10 ml, Thermo Fisher Scientific) and washed for 20 min with amylose wash buffer using gravity flow. The last wash was carried out with amylose wash buffer containing 250 mM NaCl. Elution was carried out with amylose elution

buffer containing 250 mM NaCl and 10 mM maltose (Sigma), and the total protein concentration was estimated by the Bradford assay (Bio-Rad). To cleave off the maltose binding tag (MBP), 1/5 (weight/weight) of PreScission protease (PP) with respect to the total protein in the eluate was added and incubated for 2 h at 4 °C[66]. Next, the cleaved amylose eluate was diluted by adding dilution buffer (50 mM Tris·HCl, pH 7.5, 10% glycerol, 5 mM β-ME), in order to bring NaCl concentration to 100 mM. The diluted eluate was then applied on a 5 ml heparin chromatography column (Cytiva) attached to an ÄKTA pure system (Cytiva), equilibrated in Buffer A (50 mM Tris·HCl pH 7.5, 5 mM β-ME, 10% glycerol, 100 mM NaCl). The column was washed with Buffer A, and the protein was eluted in fractions during a gradient 0–100% of Buffer B (50 mM Tris·HCl, pH 7.5, 5 mM β-ME, 10% glycerol, 1 M NaCl). Fractions were pooled, aliquoted, snap-frozen and stored at − 80 °C.

Human Polδ was expressed in Sf9 cells using pSJ20-FLAG-POLD1, pSJ21-POLD2-POLD3-POLD4 vectors[67]. Sf9 cells were seeded at 5,000,000 cells/ml in 500 ml 18 h before infection. The cells were then co-infected with corresponding baculoviruses and incubated for 52 h at 27 °C with constant agitation. The cells were harvested by centrifugation (500 × g, 10 min) and washed once with PBS. The pellets were snap-frozen in liquid nitrogen and stored at − 80 °C. All subsequent steps were carried out at 4 °C. The pellet was resuspended into 3 volumes of lysis buffer containing 50 mM Tris·HCl pH 7.5, 0.5 mM β-ME, 1 mM EDTA, 1 mM PMSF, 1:300 (volume/volume) protease inhibitor cocktail (Sigma), 60 μg/ml leupeptin (Merck), 1 tablet of complete EDTA-free protease inhibitor (Roche), 1 μg/ml Pepstatin (Sigma), 0.5 μg/ml Aprotinin (Sigma), and incubated for 20 min while stirring. Glycerol was added to a final concentration of 10%, NaCl was added to a final concentration of 305 mM. The suspension was further incubated for 30 min and centrifuged at 55,000 × g for 30 min to obtain soluble extract. The cleared extract was incubated for 1 h with pre-equilibrated FLAG-conjugated M2 agarose (Sigma). The resin was collected by centrifugation at 2000 × g for 2 min, followed by one wash batchwise with wash buffer (50 mM Tris·HCl, pH 7.5, 0.5 mM β-ME, 300 mM NaCl, 10% glycerol, 1 mM PMSF, 0.1% nonyl phenoxypolyethoxylethanol [NP-40, Sigma]). Upon being transferred to a 5 ml disposable column (Thermo Fisher Scientific), the resin was washed for 30 min with wash buffer. Elution was carried out with FLAG elution buffer (50 mM Tris·HCl, pH 7.5, 0.5 mM β-ME, 100 mM NaCl, 10% glycerol, 1 mM PMSF, 200 μg/ml 3 x FLAG peptide [GlpBio]), in fractions, and the total protein concentration was estimated using the Bradford assay (Bio-Rad). Protein-containing fractions were pooled, and β-ME was added to reach the final concentration of 2 mM. The eluate was applied on a 1 ml HiTrap Q chromatography column (Cytiva) attached to an ÄKTA pure system (Cytiva), previously equilibrated in equilibration buffer (50 mM Tris·HCl pH 7.5, 2 mM β-ME, 10% glycerol, 50 mM NaCl). The column was washed with equilibration buffer containing 150 mM NaCl and eluted in fractions in a gradient 0-100% of Buffer B (50 mM Tris·HCl pH 7.5, 2 mM β-ME, 10% glycerol, 1 M NaCl). Fractions were pooled, aliquoted, snap-frozen and stored at − 80 °C. Yeast RFC and PCNA were expressed in E. coli and purified by chromatography[23,68,69]. Yeast polδ was expressed in the yeast strain WDH668 and purified following existing protocols[57].

The sequence for the expression of 3xFLAG-Pif1 (aa 40–859) was codon-optimized for the expression in Saccharomyces cerevisiae and obtained from GenScript. The gene was then cloned into pJF2-pRS303/Ctd1-Gal1-Gal10-Gal4 (a kind gift from John Diffley, Crick Institute) to generate the pJF2-pRS303yPif1coy-Gal1-10-Gal4 plasmid. ScPif1 sequence was PCR-amplified from the pJF2-pRS303yPif1coy-Gal1-10-Gal4 plasmid using the following primers: Fw-MBP-yPif1-His-insect, Rv-MBP-yPif1-His-insect (Supplementary Table 1). The amplified product was inserted into the pFastBac-MBP-hPIF1co-10xHis vector[70] using the restriction sites NheI and MluI to generate pFastBac-MBP-ScPif1-10xHis. ScPif1 was expressed in Spodoptera frugiperda 9 (Sf9) cells using the Bac-to-Bac expression system (Thermo Fisher Scientific),

according to the manufacturer's recommendations. Frozen Sf9 cells pellets from 500 ml cultures were resuspended in lysis buffer (50 mM Tris·HCl, pH 7.5, 5 mM β-mercaptoethanol, 1 mM PMSF, 1 mM EDTA, protease inhibitor cocktail [P8340, Sigma-Aldrich] diluted 1:300, 30 μg/ml leupeptin [Merck Millipore]) and incubated at 4 °C for 20 min. Glycerol was added to a final concentration of 16.7%, NaCl was added to a final concentration of 305 mM and the solution was incubated at 4 °C for 30 min. The mixture was centrifuged at 55,000 × g at 4 °C for 30 min. The resulting soluble extract was incubated with 4 ml amylose resin (New England Biolabs) at 4 °C for 1 h. The resin was washed with amylose wash buffer 1 M (50 mM Tris·HCl pH 7.5, 2 mM β-mercaptoethanol, 1 mM PMSF, 10% glycerol, 1 M NaCl) followed by amylose wash buffer 300 mM (50 mM Tris·HCl pH 7.5, 2 mM β-mercaptoethanol, 1 mM PMSF, 10% glycerol, 300 mM NaCl). Proteins were eluted using amylose elution buffer (50 mM Tris·HCl, pH 7.5, 2 mM β-mercaptoethanol, 1 mM PMSF, 10% glycerol, 300 mM NaCl, 10 mM maltose [Sigma]). The MBP-tagged protein was incubated with Pre-Scission protease (~ 20 μg PreScission protease per 100 μg of tagged protein, 1:5 ratio) at 4 °C for 1 h to cleave the MBP tag. After cleavage, the protein was supplemented with Imidazole (Sigma) to a final concentration of 10 mM and applied onto pre-equilibrated Ni-NTA agarose resin (Qiagen) at 4 °C in flow. The resin was washed with Ni-NTA buffer 1 (50 mM Tris·HCl pH 7.5, 5 mM β-mercaptoethanol, 10% glycerol, 1 mM PMSF, 30 mM Imidazole and 1 M NaCl) and subsequently with Ni-NTA buffer 2 (50 mM Tris·HCl pH 7.5, 5 mM β-mercaptoethanol, 150 mM NaCl, 10% glycerol, 1 mM PMSF, 30 mM imidazole). The protein was eluted with Ni-NTA elution buffer (50 mM Tris·HCl, pH 7.5, 5 mM β-mercaptoethanol, 150 mM NaCl, 10% glycerol, 1 mM PMSF, 300 mM imidazole). Fractions with high protein concentration, as estimated by the Bradford assay, were pooled and dialyzed for 1 h at 4 °C against 1 L of buffer containing 50 mM Tris·HCl pH 7.5, 5 mM β-mercaptoethanol, 150 mM NaCl, 10% glycerol, 0.5 mM PMSF and then snap-frozen in liquid nitrogen. Human DNA2 protein was expressed in Sf9 cells and purified following existing protocols.

## Preparation of plasmid-based DNA substrates

The plasmid pRichi is a derivative of pGEM-13Zf(+)[71]. pRichi was further modified by cloning in a sequence containing seven tandem nickase sites (Nt.BbvCI/Nb.BbvCI) interrupted by a BamHI sequence in between NotI and HindIII restriction sites, using oligonucleotides listed in Supplementary Table 2, to create pSam. The full sequence of pSam is listed in Supplementary Table 3. To construct pSam_Blue and pSam_Red (Fig. 2a, b), the sequence between NotI and AatII sites in the original pSam backbone was replaced with two different unrelated sequences of the same length (see Supplementary Table 3). To insert various structures, 10 μg of pSam was digested with 10 U of either Nt.BbvCI or Nb.BbvCI (New England Biolabs) for 2 h at 37 °C in rCutSmart buffer (New England Biolabs). The digested product was then heated to 85 °C for 10 min and immediately purified using a QIAquick spin column (Qiagen). The column recovers approximately 80% of material when using 10 μg DNA. The resulting gapped plasmid was annealed with an oligonucleotide containing the desired structure at a gap-to-oligonucleotide ratio of 1:7 in rCutSmart buffer (New England Biolabs) by decreasing the temperature from 85 °C to 4 °C at a rate of 0.01 °C/s in a PCR thermocycler. The sequences of the oligonucleotides used for substrate preparation are listed in Supplementary Table 4. The annealed product was ligated with 10 U of T4 DNA ligase (Thermo Fisher Scientific) at 37 °C for 2 h. The ligase was heat-inactivated at 70 °C for 5 min, after which the reaction products were treated with 10 U of T5 exonuclease (New England Biolabs) to remove unligated DNA intermediates, and 20 U BamHI-HF (New England Biolabs) to remove the original DNA sequence. See Supplementary Fig. 2a for a cartoon of substrate DNA preparation. The final DNA substrate was purified using a QIAquick spin column (Qiagen) and analyzed on a 1% agarose gel.

## Preparation of oligonucleotide-based DNA substrates

The sequences of all oligonucleotides used for DNA substrate preparation are listed in Supplementary Table 4. Oligonucleotides were $^{32}$P-labeled either at the 3' terminus using [$\alpha$-$^{32}$P]dCTP (Hartmann Analytic) and Terminal Transferase (New England Biolabs) or at the 5' terminus using [$\gamma$-$^{32}$P]ATP (Hartmann Analytic) and T4 PNK (New England Biolabs), as indicated, following the manufacturer's instructions. Unincorporated nucleotides were removed using Micro-Bio P30 Spin column (Bio-Rad) according to the manufacturer's recommendations. The oligonucleotide-based (CAG)$_4$ structured DNA substrate was prepared using the following oligonucleotides: CAG_X4_top-strand_biotin and CAG_X4_bottom_strand_biotin. The oligonucleotide-based substrate with a pre-existing nick 3 nucleotides downstream of the (CAG)$_4$ loop was prepared using the oligonucleotides Top-strand_3nt_nicked_left_part, Nick_right_part, and CAG_X4_bottom_strand_biotin. The fully duplex DNA substrate was prepared using the following oligonucleotides: Top strand oligo WT FRP and CAG_X4_bottom_strand_biotin. The nicked duplex DNA substrate was prepared using the oligonucleotides Top strand oligo right part WT FRP, Top strand oligo left part WT FRP and CAG_X4_bottom_strand_biotin.

## dCas9 RNP generation

crRNA (CD.Cas9.GJXV6830.AD) and tracrRNA for Cas9 targeting were purchased from Integrated DNA Technologies (see Supplementary Table 5 for oligonucleotide sequences) and annealed at equimolar concentrations (10 μM final) in IDTE buffer pH 8.0 (Integrated DNA Technologies) according to the manufacturer's instructions. 1 μM (final) dCas9 (a kind gift from Martin Jinek, University of Zurich)[70], was incubated with a three-fold excess of the RNA component (annealed crRNA·tracrRNA) in RNP buffer containing 25 mM Tris-HCl pH 7.5, 5 mM MgCl$_2$, 1 mM DTT, 0.25 mg/ml recombinant albumin (New England Biolabs) and 150 mM KCl for 10 min at 25 °C to produce the respective RNPs. After incubation, the RNPs were subaliquoted, snap-frozen in liquid nitrogen, and stored at − 80 °C for later use.

## Southern blot-based nuclease assays

Reactions (15 μl final volume) were performed in 25 mM Tris-acetate pH 7.5, 2 mM ATP, 5 mM MgCl$_2$, 0.1 mg/ml recombinant albumin (New England Biolabs), 1 mM DTT and 100 ng of pSam plasmid DNA with either (T)$_4$ or (CAG)$_4$ extrahelical loops. Reactions were assembled on ice as described in the figures. The protein concentrations were: 75 nM MutLγ (or MutLγ 3ND), 75 nM MutLα, 75 nM FAN1 and its variants, 75 nM MutSβ, MutSα, 100 nM PCNA, 100 nM RFC and 50 nM EXO1 D173A, unless indicated otherwise. All the reactions were performed without added NaCl or KCl; the only salt present in the reaction was the residual NaCl carried over with the recombinant protein storage buffer, resulting in a final salt concentration about 40 mM. Reactions were supplemented with protein storage buffer to equalize the final NaCl concentration across all samples. We note that the MutLγ nuclease is sensitive to salt[23]. For the dCas9 reactions, the RNP containing the target sequence (see Supplementary Table 5) was first incubated at 37 °C for 30 min with the substrate before the proteins were added. Reactions were incubated for 1 h at 37 °C, followed by incubation at 78 °C for 10 min. DNA was then linearized using ScaI (New England Biolabs) according to the manufacturer's recommendations. The reactions were terminated by adding 5 μl of STOP solution (150 mM EDTA, 0.2% SDS, 30% glycerol, 0.01% bromophenol blue) and 1 μl of Proteinase K (Roche, 18 mg/ml). The mixture was further incubated at 50 °C for 1 h. The digested products were denatured by boiling for 5 min at 95 °C in the presence of 500 mM NaOH and 10 mM EDTA. Denatured DNA was resolved on a 1% alkaline agarose gel in 50 mM NaOH and 1 mM EDTA for 20 h at 1 V/cm. The separated DNA was transferred onto a Hybond N+ membrane (Cytiva) and subjected to Southern hybridization using indicated oligonucleotide probes $^{32}$P-labeled at both 5' and 3' termini (1.2 nM final). All used

oligonucleotide-based probes are listed in Supplementary Table 6. Membranes were washed, exposed to storage phosphor screens (Cytiva), and scanned using a Typhoon Phosphor Imager FLA 9500 (GE Healthcare). Signals were quantified using ImageJ2 (NIH) and plotted with Prism 10 (GraphPad).

## Oligonucleotide-based nuclease assays

Nuclease assays were performed in a 15 μl reaction volume containing 25 mM Tris-acetate, pH 7.5, 5 mM MgCl$_2$, 2 mM ATP, 1 mM DTT, and 0.25 mg/ml recombinant albumin (New England Biolabs), with 1 nM DNA substrate (in molecules). All the reactions were performed without added NaCl or KCl; the only salt present in the reaction was the residual NaCl carried over with the recombinant protein storage buffer, resulting in a final salt concentration about 40 mM. Reactions were supplemented with protein storage buffer to equalize the final NaCl concentration across all samples. To block both DNA ends, 15 nM monovalent streptavidin (M. Howarth, University of Oxford) was added, and the samples were incubated at room temperature for 5 min. Unless indicated otherwise, 25 nM FAN1 (or FAN1 ND), 30 nM RFC (or yRFC), 30 nM PCNA (or yPCNA), and 30 nM Polδ (or yPolδ) were added on ice, and the reactions were incubated at 37 °C for 30 min. Where indicated, dNTPs were added to a final concentration of 100 μM. Reactions were terminated by adding 0.5 μl of 0.5 M EDTA and 1 μl of Proteinase K (Roche, 18 mg/ml), followed by incubation at 50 °C for 30 min. An equal volume of formamide dye (95% [volume/volume] formamide, 20 mM EDTA, bromophenol blue) was added, and the samples were heated at 95 °C for 5 min. The reaction products were separated on 20% denaturing polyacrylamide gels (acrylamide:bisacrylamide ratio 19:1, Bio-Rad). After electrophoresis, gels were fixed in a solution of 40% methanol, 10% acetic acid, and 5% glycerol for 30 min, dried on 3 MM paper (Whatman), and exposed to storage phosphor screens (Cytiva). The exposed screens were scanned using a Typhoon 9500 Phosphor Imager (Cytiva). Signals were quantified using ImageJ2 (NIH) and plotted with Prism 10 (GraphPad).

## Nicking assays with plasmid-length DNA

Reactions (15 μl final volume) were performed in a buffer containing 25 mM Tris-acetate, pH 7.5, 1 mM DTT, 0.1 mg/ml recombinant albumin (New England Biolabs), 5 mM MgCl$_2$, and 2 mM ATP. Each reaction included 100 ng of plasmid-based DNA substrate, either 5.6 kbp-long pFB-RPA2 or 3.2 kbp-long pSam (CAG)$_4$, as indicated. All the reactions were performed without added NaCl or KCl; the only salt present in the reaction was the residual NaCl carried over with the recombinant protein storage buffer, resulting in a final salt concentration about 40 mM. Reactions were supplemented with protein storage buffer to equalize the final NaCl concentration across all samples. Where indicated, ATP was substituted with ATP-γ-S (Cayman). Next, 75 nM MutLγ (or MutLγ 3ND), 75 nM MutSβ, 75 nM MutSγ, 75 nM RFC (or yRFC), 75 nM PCNA (or yPCNA), 50 nM EXO1 D173A and 50 nM FAN1 (or FAN1 ND) were added on ice and, unless specified otherwise, the reactions were incubated at 37 °C for 1 h. Reactions were stopped by adding 5 μl of STOP solution (150 mM EDTA, 2% SDS, 30% glycerol, 0.01% bromophenol blue) and 1 μl of Proteinase K (Roche, 18 mg/ml), followed by incubation at 50 °C for 1 h. The resulting products were analyzed via electrophoresis on 1% agarose gels containing GelRed (Biotium) and run in TAE buffer (40 mM Tris, 20 mM acetate and 1 mM EDTA) for 90 min at 6 V/cm. The gels were then imaged using a Quantum gel imager (Vilber). Signals were quantified using ImageJ2 (NIH) and plotted with Prism 10 (GraphPad).

## DNA strand displacement assays

Reactions (15 μl) were performed in a buffer containing 25 mM Tris-acetate, pH 7.5, 2 mM ATP, 5 mM MgCl$_2$, 1 mM DTT, 0.1 mg/ml recombinant albumin (New England Biolabs) and 100 ng of DNA substrate. 75 nM MutLγ and 75 nM MutSβ were added, and the samples

were incubated for 1 h at 37 °C. The reactions were then incubated with 40 nM Polδ, 40 nM RFC, 40 nM PCNA for 5 min at 37 °C. Subsequently, 100 μM dNTPs, 80 nCi/μl [α-³²P]dCTP (Hartmann Analytic) and 82 nM RPA were added, and the reactions were incubated for an additional 30 min at 37 °C. For the yeast experiment, 30 nM yPolδ alone or with 30 nM yRFC, 30 nM yPCNA were incubated 5 min at either 30 °C or 37 °C. Subsequently, 100 μM dNTPs, 80 nCi/μl [α-³²P]dCTP (Hartmann Analytic), 100 nM yRPA, and 50 nM of yPif1 were added, and the reactions were incubated for an additional 30 min at either 30 °C or 37 °C. The reactions were stopped by adding 5 μl of STOP solution (150 mM EDTA, 0.2% SDS, 30% glycerol, 0.01% bromophenol blue) and 1 μl of Proteinase K (Roche, 18 mg/ml). The samples were then incubated at 50 °C for 30 min. Reaction products were analyzed by electrophoresis on a 1% agarose gel at 4 V/cm for 3 h. The gels were dried on DE81 chromatography paper (Whatman) and analyzed as described above for the oligonucletide-based nuclease assays. Signals were quantified using ImageJ2 (NIH) and plotted with Prism 10 (GraphPad).

## GLOE-Seq

The protocol was adapted from the one described by Petrosino et al.[48]. All the oligonucleotides listed below were used and named as reported[48]. The nuclease reactions with the indicated proteins and the pSAM_(CAG)₄ substrate were performed as described above for the nicking assays in 15 μl volume. The nuclease reactions were terminated by incubation at 95 °C for 10 min, followed by 5 min on ice. A ligation step was then performed to capture the free 3′-OH termini as follows: 14.85 nM DNA, 6.5 μl of 10 x T4 DNA ligase Buffer (Thermo Fisher Scientific), 1.4 μl of 5 μM stock proximal adapter purchased from Integrated DNA Technologies (7.5 x excess), 19.5 μl of 50% PEG 8000 (Thermo Fisher Scientific), 3 μl of T4 DNA ligase (Thermo Fisher Scientific), and water were added to reach a final volume of 65 μl. The PCR was set up with the following conditions: annealing at 25 °C for 1 h and ligation at 22 °C for 2 h. The ligated product was purified using AMPure beads (Beckman Coulter) and eluted in 103 μl of water. The purified product was then sonicated using a Bioruptor Pico (20 cycles of 30 sec on/30 sec off). The sonicated product was purified twice with AMPure beads and once with streptavidin MyOne C1 Dynabeads (Thermo Fisher Scientific). The beads were washed once with 1 x SSC buffer (150 mM NaCl, 15 mM sodium citrate, pH 7.0) for 5 min on a rotating wheel, followed by one wash with 20 mM NaOH for 10 min on a rotating wheel. DNA was eluted by adding 16 μl of H₂O. For second-strand DNA synthesis, the following reaction was performed in a PCR tube: 14.85 μl of eluate, 1.5 μl of oligonucleotide HU3790 purchased from Integrated DNA Technologies, 0.3 unit of Phusion polymerase, 6 μl of 5 x HF buffer (New England Biolabs; 200 mM Tris-HCl pH 7.4, 100 mM KCl, 100 mM (NH₄)₂SO₄, 20 mM MgCl₂, 1% Triton X-100), 5 μl of H₂O, and 0.6 μl of 10 μM dNTP mix. The reaction was incubated in a PCR machine using the following program: denaturation at 95 °C for 2 min, annealing at 60 °C for 30 sec, and extension at 72 °C for 2 min. The amplified dsDNA was purified using AMPure beads (Beckman Coulter), followed by a polishing reaction performed with the NEBNext Ultra II DNA Library Prep Kit (New England Biolabs) with the following PCR conditions: 98 °C 30 min; 98 °C 10 min; 65 °C 1 min 15 sec; 65 °C 5 min; 4 °C forever. The first 3 steps were repeated for a total of 12 cycles. The product was then ligated with the distal adapter purchased from Integrated DNA Technologies using the reagents from NEBNext Ultra II DNA Library Prep Kit (New England Biolabs) for 20 min at 20 °C. The product was purified using AMPure beads and eluted in 20 μl H₂O. Finally, the DNA library was amplified following the NEBNext Ultra II DNA Library Prep Kit (New England Biolabs) and sequenced using Nextseq2000 with settings for single-end 120 nt reads and 8 + 8 indices.

The obtained sequences were aligned to the corresponding reference plasmids using HISAT v2.1.0[72]. The reference sequence was duplicated and concatenated three times to consider the plasmid's

circular nature. Subsequently, the duplicated sequence was selected, and a custom Python script (https://github.com/simonemoro/cutFinder) was employed to determine the strand-specific cleavage events. Reads aligned to the top strand were used to infer cuts occurred on the bottom strand, and vice versa. Next, the unmapped reads were extracted and subjected to an additional mapping step against the corresponding reference sequence, specifically targeting the loop-containing strand while excluding the loop itself. This approach was implemented to account for the reads that would otherwise remain undetected in the analysis. The data resulting from the analysis, which contains the sequence and the mapped reads, were plotted in a polar plot with a custom Python script for visualization.

## Electrophoretic mobility shift assays

The MutSβ DNA binding reactions were carried out in 15 μl volume in a buffer containing 25 mM Tris-acetate, pH 7.5, 1 mM DTT, 2 mM ATP, 5 mM MgCl₂, 0.1 mg/ml recombinant albumin (NEB), 0.5 nM DNA substrate (in molecules) and the indicated concentrations of recombinant MutSβ. The reactions were supplemented with a competitor plasmid dsDNA (3 ng/μl of pUC19). The reactions were assembled on ice, incubated at 37 °C for 30 min, followed by the addition of 5 μl EMSA loading dye (50% glycerol, 0,01% bromophenol blue). The products were separated on a native 6% polyacrylamide gel. Gels were dried on 17CHR chromatography paper (Whatman), exposed to storage phosphor screens (Cytiva), and scanned using the Typhoon Phosphor Imager FLA 9500 (Cytiva).

## Protein interaction assays

To test the interaction between MutLγ and FAN1, 0.6 μg anti-MLH1 antibody (Abcam, ab223844) were immobilized on 10 μl protein G magnetic beads (Dynabeads, Thermo Fisher Scientific). The antibody was incubated with the beads in 50 μl of PBS-T (PBS with 0.1% Tween-20) at 4 °C for 1 h with gentle mixing. The excess of unbound antibody was removed by washing the beads twice with 150 μl of PBS-T. The antibody-coated beads were then incubated with 1 μg of recombinant MLH1-MLH3, in 60 μl immunoprecipitation buffer (25 mM Tris-HCl, pH 7.5, 3 mM EDTA, 1 mM DTT, 0.2 μg/μl recombinant albumin [New England Biolabs], 70 mM NaCl) at 4 °C for 1 h with gentle agitation. Upon incubation with the bait protein, the beads were washed three times with 150 μl wash buffer (50 mM Tris-HCl, pH 7.5, 3 mM EDTA, 1 mM DTT, 0.1% Triton X-100, 100 mM NaCl). Subsequently, 1 μg of recombinant MSH2-MSH3, recombinant MBP-FAN1, recombinant MBP-FAN1-MIP-MIM or MSH2-MSH3 followed by MBP-FAN1 or recombinant MBP-FAN1-MIP-MIM were added to the beads in 70 μl of immunoprecipitation buffer. The samples where MSH2-MSH3 or MBP-FAN1 was added individually were incubated at 4 °C for 1 h with gentle agitation. The samples where MSH2-MSH3 was added in combination with MBP-FAN1 or MBP-FAN1-MIP-MIM were incubated first with 1 μg MSH2-MSH3 for 20 min with gentle agitation, followed by the addition of 1 μg MBP-FAN1 or MBP-FAN1-MIP-MIM to reach a total incubation time of 1 h. The beads were washed three times with 150 μl wash buffer, and bound proteins were eluted by boiling the beads in SDS buffer (50 mM Tris-HCl, pH 6.8, 1.6% SDS, 100 mM DTT, 10% glycerol, 0.01% bromophenol blue) for 3 min at 95 °C. The eluted proteins were resolved on a 10% SDS-PAGE gel and analyzed by western blotting using anti-MBP (1:500, MBL, M091-3) and anti-FLAG (1:1000, Sigma, F3165) antibodies.

To test the interaction between FAN1, and RFC and PCNA, soluble extract containing MBP-FAN1 was prepared as during protein purification. The soluble extract (200 μl) was incubated with 60 μl of amylose resin (New England Biolabs) at 4 °C for 1 h with gentle rotation. The excess of unbound protein was removed by washing the resin three times with Wash buffer A (50 mM Tris-HCl, pH 7.5, 2 mM EDTA, 2 mM β-ME, 0.05% NP-40, 1 mM PMSF, 300 mM NaCl), and twice with Wash buffer B (50 mM Tris-HCl pH 7.5, 2 mM EDTA, 2 mM β-ME, 0.05%

NP-40, 1 mM PMSF, 80 mM NaCl). The resin was then incubated with 1.5 μg of recombinant RFC, 1.5 μg of recombinant PCNA, or both, in 200 μl Wash buffer B, at 4 °C for 1 h with gentle agitation. Upon incubation with the bait protein, the resin was washed three times with 500 μl Wash buffer B, and bound proteins were eluted in 100 μl Wash buffer B containing 20 mM maltose, on ice for 10 min, with occasional tapping. 16 μl of eluate were mixed with 4 μl of SDS loading buffer (50 mM Tris-HCl, pH 6.8, 1.6% SDS, 100 mM DTT, 10% glycerol, 0.01% bromophenol blue) and heated for 5 min at 95 °C. The eluted proteins were resolved on a 10% SDS-PAGE gel and analyzed by western blotting using the following antibodies: anti-FAN1 (1:500, ab68572, Abcam), anti-RFC3 (1:500, sc-390293, Santa Cruz Biotechnology), and anti-6x His Tag (1:1000, PA-983B, Invitrogen) for the detection of His-tagged PCNA.

### Repeat contraction and expansion assays analyzed by Sanger/TIDER

Reactions (15 μl total volume) were performed in a buffer containing 25 mM Tris-acetate, pH 7.5, 2 mM ATP, 5 mM MgCl$_2$, 1 mM DTT, 0.1 mg/ml recombinant albumin (New England Biolabs) and 100 ng of pSAM_(CAG)$_4$ DNA substrate. 75 nM MutLγ and 75 nM MutSβ were added together first, and samples were incubated for 1 h at 37 °C. The reactions were then incubated with 40 nM Polδ, 75 nM RFC and 75 nM PCNA for 5 min at 37 °C. Subsequently, 100 μM dNTPs, 80 nCi/μl [α-$^{32}$P] dCTP (Hartmann Analytic) and 82 nM RPA were added, and the reactions were incubated for an additional 30 min at 37 °C. When indicated, 75 nM DNA2 was added in the mix and incubated for 30 min at 37 °C. In the alternative setup, 75 nM FAN1, 75 nM RFC, 75 nM PCNA, 75 nM MutLγ, and 75 nM MutSβ were combined and incubated for 1 h at 37 °C. 40 nM Polδ was then added for 5 min at 37 °C. This was followed by the addition of 100 μM dNTPs, 80 nCi/μl [α-$^{32}$P]dCTP (Hartmann Analytic) and 82 nM RPA, and the samples were incubated further for 30 min at 37 °C. The reactions were stopped by adding 5 μl of STOP solution (150 mM EDTA, 0.2% SDS, 30% glycerol, 0.01% bromophenol blue) and 1 μl of Proteinase K (Roche, 18 mg/ml). The samples were then incubated at 50 °C for 30 min and purified on a MinElute column (Qiagen). Purified reaction products were analyzed by Sanger sequencing (Macrogen) using primers listed in Supplementary Table 7. To quantify and analyze insertions and deletions in the plasmid DNA, we used TIDER, a tool previously employed for evaluating CRISPR/Cas9-induced mutations[50]. In our case, no CRISPR/Cas9 system was used, and thus, we did not have a standard reference chromatogram for comparison. To overcome this, a newly developed TIDER feature was applied. Instead of relying on an existing reference chromatogram, this feature simulates a reference by modifying the control sequence itself. It introduces or removes peaks to mimic the expected insertions, deletions, or substitutions at the target site, effectively replicating the kind of DNA changes typically seen in CRISPR experiments. By generating this synthetic reference, TIDER can accurately identify and quantify indels, even in the absence of a true reference chromatogram.

### Statistical analysis and reproducibility

Data are presented as mean ± SEM, with the number of independent experiments (n) indicated in each figure legend. In Fig. 4f–h, the experiments were repeated three times. For comparisons between two conditions, unpaired two-tailed Student's *t* tests were used. Datasets involving more than two groups under a single variable were analyzed by ordinary one-way ANOVA followed by Tukey's multiple comparisons test to control the family-wise error rate. For experiments with two independent factors (protein components and looped vs. non-looped DNA strand), a full two-way ANOVA including the interaction term was carried out, and Tukey's post-hoc tests were used to probe simple effects within each row and column while controlling the family-wise error rate. All statistical analyses were conducted in GraphPad Prism 10.

### Reporting summary

Further information on research design is available in the Nature Portfolio Reporting Summary linked to this article.

## Data availability

The raw sequencing data from the GLOE-seq have been deposited in the European Nucleotide Archive (https://www.ebi.ac.uk/ena/browser/home) under the study accession number PRJEB91651. Source data are provided in this paper.

## Code availability

The code used for the GLOE-seq analysis can be found at GitHub (https://github.com/simonemoro/cutFinder).

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

## Acknowledgements

Research in the Cejka laboratory is funded by the Swiss National Science Foundation (SNSF) (Grants 310030_207588 and 310030_205199) and the European Research Council (ERC) (Grant 101018257). V.M. was funded by a grant from Swiss Cancer Research (KFS-5397-08-2021). I.S. and V.M. were partly funded by the CHDI Foundation. We thank Josef Jiricny (ETH Zurich) for discussions and plasmid DNA, and Martin Jinek (University of Zurich) for dCas9.

## Author contributions

I.S. expressed and purified most recombinant proteins, prepared all DNA substrates and performed and analyzed almost all biochemical experiments under the supervision of P.C. V.M. performed the DNA nicking and protein interaction assays with FAN1. A.R. and A.C. performed the NGS part for the GLOE-seq procedure. A.M. carried out the TIDER analyses. G.R. and S.M. helped with GLOE-seq analysis. A.A. prepared recombinant Polδ protein. E.C. prepared wild type FAN1. I.C. helped prepare FAN1 MIP-MIM ND. M.R. prepared EXO1 variants. A.J. prepared yeast Pif1. I.S. and P.C. wrote the manuscript with inputs from all authors.

## Competing interests

A.M. is an employee of Data Curators B.V. The remaining authors declare no competing interests.
