## [Transparent Peer Review file · Nature Communications]

Mechanism of trinucleotide repeat expansion by MutS β -MutL γ and contraction by FAN1

Corresponding Author: Professor Petr Cejka

Version 0:

Reviewer comments:

Reviewer #1

(Remarks to the Author)

This is an exciting biochemical study that sheds new light on the mechanisms of repeat expansions and contractions. I shall add right away that this is the most clear, beautiful biochemistry that I've seen in years!

The study has two major results. The authors used double-stranded DNA substrates containing a (CAG)₄ extrahelical loop in one of its strands. They convincingly demonstrated that MutS β -MutL γ complex efficiently cleaves the strand opposite to this loop. RFC-loaded PCNA has a moderate inhibitory effect on the efficiency of the cleavage but restricts it to a site 5' to the loop on the opposite strand. This incision site was shown to serve as a primer for DNA synthesis by Pol δ (in conjunction with RPA, RFC and PCNA), which ultimately leads to the insertion of the (CTG)₄ sequence complementary to the extrahelical loop sequence, i.e. repeat expansion equivalent.

The second major result is that FAN1 nuclease - which is known to counteract repeat expansions - cleaves covalently closed DNA 3' to the extrahelical loop followed by exonucleolytic DNA degradation. In the presence of RFC and PCNA, the endonuclease activity of FAN1 was enhanced, while its exonuclease activity was suppressed. When Pol δ was added to the reaction, its exonuclease activity excised the loop, and subsequent resynthesis created DNA strand without the repeat, i.e. repeat contraction.

Altogether, this is a convincing demonstration of repeat expansions in vitro triggered by MutS β -MutL γ complex versus repeat contractions triggered by the FAN1 nuclease.

I have just two comments with regard to biology of the matter. First, in the author's system, an elementary step of repeat expansion is limited to the loop-out size. Translating this observation into humans might explain the late onset of most repeat-expansion diseases, as many consecutive rounds of the loop-out addition would be needed to reach the disease threshold. The authors may want to expand on this point. Second, I very much like the model of larger repeat expansions in proliferating cells (Fig. 6b), which are driven by nicks, subsequent replication fork collapse and DSB repair by HR. Notably, this exact sequel of events was recently demonstrated in the paper of Li et al (PMID: 39585990), which the authors forgot to cite.

Reviewer #2

(Remarks to the Author)

1. Noteworthy results

The authors provide convincing and detailed mechanistic insight into how three key protein complexes/proteins - MutS β , MutL γ , and FAN1 - may contribute to trinucleotide repeat (TNR) instability

They reproduced some published data by other groups, especially around MutL γ nicking the non-nicked strand and FAN1 nicking the looped strand. They however provide novel insights, arguing that FAN1 does not need a pre-existing nick to cut the looped strand as previously suggested (Phadte et al). Moreover, they uncovered a sequence preference for MutS β -activated MutL γ not previously appreciated. RFC alone or with PCNA restrict cutting by both FAN1 and MutL γ . This was substantiated by a very nice dCas9 experiment showing that it is a physical barrier. Intriguingly, RFC alone could enhance

the nuclease activity of FAN1. Importantly, they find that MutS β , MutL γ , RPA, RFC, PCNA, and Pol δ are sufficient to create expansions in vitro. They identified a potential role of FAN1 and Pol δ in creating contractions.

2. Will the work be of significance to the field and related fields? How does it compare to the established literature? If the work is not original, please provide relevant references.

The research presented in this paper will be of great interest in the field, building upon supporting data from years of work in experimental models and genome-wide association studies to explain the oppositional roles for MutL and FAN1 at the DNA strand and nucleotide resolution. The work strongly supports ongoing translational research by other groups targeting MSH3 (MutS β), MLH3 (MutL γ), and FAN1 and may inform targeting of new epitopes or interactions.

The data presented also explain previous genetic reports of FAN1 competition with MutL at the nuclease and structural levels. The authors greatly expand on the established literature, using very well-controlled experiments and innovative techniques (e.g., GLOE-Seq) to improve understanding of the events leading to MutS β -MutL γ and FAN1-mediated repeat instability, whilst also defining novel roles for additional key proteins.

3. Does the work support the conclusions and claims, or is additional evidence needed?

The work does generally meet the conclusions and claims. I would like to highlight the numerous controls done on how the purified proteins are active on control substrates, for example, and how yeast proteins cannot substitute for human ones. Overall, the experiments are performed at a very high standard.

However, some additional statistical evidence would support the results further:

- Where Southern blots have been quantified and graphed, p-values and significant differences should be included on all graphs (ie. Fig. 1i, Fig. 4c)
- All Southern blots should be quantified, with statistical results detailed in the supplementary data, if not in main text.
- Clarification on how data were statistically analysed – if t-tests were used, were all data individually compared or select data only? If select, then which and why? Would an ANOVA be better in much of these cases?
- A statistical analysis section is missing from the methods

4. Are there any flaws in the data analysis, interpretation and conclusions? - Do these prohibit publication or require revision?

Beyond the clarification on the statistical analysis above, the manuscript can be published as is. I have, however, some minor comments that would be good to address.

It would be good to specify why only CAG loops were used and not CTG ones. I would not want to have the authors redo the whole of the experiments with CTG loops as they have not made claims about CTG loops and it would double the size of the paper.

How were experimental conditions (e.g., protein concentration and buffer conditions) decided upon? Were these physiological conditions or were there trials of different conditions?

i. Supplementary Figure 1. However, can MutS β stimulate MutL γ DNA cleavage at DNA substrates with CAG loops? As this would have an impact on the necessity of MutS β (and the therapeutic targeting of MSH3) in MutL γ -mediated loop incorporation and repeat expansion.

ii. This is beyond the scope of the study and only for the authors to think about. One outstanding question from the genetics is that all three MutL complexes are essential for repeat expansion. How they would interact and indeed whether they process this loop out substrate separately would be a very interesting question to answer using the approaches herein.

iii. Figure 2a: There is a distinct band in lane 4 (without MutS/MutL) which is only present in the other lanes with MutS and MutL. Is this cleaved DNA?

iv. "RFC directly controls FAN1, which is likely dependent on a direct interaction" (Line 191 – 192) Can the authors show direct interaction between RFC and FAN1 using pulldown or other method? Or speculate how RFC may be interacting with FAN1 in the absence of PCNA? It is difficult to reconcile how RFC may regulate FAN1 in the absence of PCNA. Specifically, what do they mean by 'loading FAN1' here?

v. PCNA only modestly increases FAN1 endonuclease activity when added to RFC (Line 189 – 194) If RFC stimulates FAN1 endonuclease activity, and addition of PCNA only modestly increased this further, could the authors speculate on discrepancies from other data? Namely the necessity for both RFC and PCNA for DNA cleavage of loop-containing DNA substrate (Phadte et al., 2023) and the impact of the FAN1-PCNA interacting p.R507H onset-hastening variant (Li et al., 2024).

vi. Were human and yeast protein interaction experiments conducted under the same conditions? As yeast and human proteins have been reported to be affected differently by different cations (Kadyrova et al., 2020). Where possible, referring to specific lanes in a figure would be very helpful when making statements in the text. The authors have done this sometimes and I found it to make reading the paper much more straightforward.

vii. Additional clarity in distinguishing between extrahelical loops and CAG4 loops would be useful. I initially thought that CAG4 extrahelical loops were used throughout the experiment, but I think the initial experiments using extrahelical loops were non-CAG loops? Some additional distinction in the text (to mirror the differences in the figures) would be helpful.

Other minor issues:

- Line 121: "repat" instead of repeat
- Line 101: no figure referenced
- Figure 1F: no Y-axis scale. Same scale as E?
- Figure 1J: DNA ladder label overlapping with blot
- Figures 1J and 1K: P2 and P1 blots/maps are in different order to 1E and 1F
- 1J: no red triangle?
- Figure 1L: red triangle difficult to distinguish from orange – I suggest changing the colour scheme.
- Figure 4 has a huge amount of data. Could this be split into two figures or sectioned off?
- Figure 4C: changing "P4" to black would make interpreting strand preference easier
- Figure 4J: dividing actual blot would be helpful, rather than dashed line below
- Figure 4K figure legend: states "rectangle" rather than triangle
- Figure 5C: bands are quite difficult to make out, would be helpful to quantify and present.
- Arrow colour difficult to distinguish
- Figure 6: Model doesn't include FAN1
- Supplementary 3B: one substrate map missing

7. Is the methodology sound? Does the work meet the expected standards in your field?

Yes, the methodology is sound and meets expected standards.

8. Is there enough detail provided in the methods for the work to be reproduced?

The methods are very detailed. An addition of how statistical analyses were done should be included.

Reviewer #3

(Remarks to the Author)

In this manuscript, Cejka and colleagues employ state-of-the-art protein biochemistry and combine elegant *in vitro* assays with genome-wide next generation sequencing methods (GLOE-Seq and TIDER analysis) to evaluate the functional interplay between two DNA repair machineries, driven by either MutLg (MLH1-MLH3) or FAN1 nuclease activities, on the fate of CAG repeat stability. Most notably, their results, highlighting competing mechanisms which either facilitate (MutLg) or inhibit (FAN1) CAG repeat expansion, are largely consistent with previous biochemical studies using purified proteins as well as cell extracts (e.g. Kadyrova et al., 2020; Phadte et al. 2023) and with recent genetic data obtained in mouse models (e.g. Mouro Pinto et al. 2024).

Their experimental work is mostly solid and overall, of high technical quality. In some instances, however, and as outlined in detailed below, additional evidence is required to support or corroborate their conclusions. There are also important information missing regarding methodology of the *in vitro* reconstituted reactions, most notably the salt concentration (it looks as if all reactions were carried out at non-physiological ionic strength) and FAN1 vs. MutSb (MSH2-MSH3) concentration (i.e. stoichiometry is a key determinant of DNA repair outcome). Otherwise, their findings presented in Figure 5 rather contradict the mechanism underlying HD pathogenesis, namely that due to higher abundance of MutSb and/or higher binding affinity of MutSb to extrahelical loops compared to FAN1 (which is commonly expressed at very low levels), CAG repeats tend to expand with time in HD brains (with both MMR and FAN1 pathways present).

In conclusion, this study not only enhances our mechanistic understanding of somatic repeat expansion, which is highly relevant for the HD field, but is also of broad interest to the entire DNA repair community. Thus, should the authors be able to appropriately address the comments, a significantly revised version of manuscript would certainly fit the scope of Nature Communications.

Major points:

1. According to the Methods description, none of the reaction buffers in the reconstituted biochemical assays do contain any monovalent chloride salts (KCl or NaCl). This raises the important question as to whether certain specific stimulatory (e.g. by RFC-PCNA on FAN1 nuclease activity) or inhibitory mechanisms (e.g. by RFC-PCNA on MutSb-MutLg) observed here only occur at non-physiological (i.e. low salt) conditions or whether they are also active at physiological ionic strength?

Considering the fact that MutLg displays DNA binding activity in absence of MutSb complexes at low salt conditions (Kadyrova et al, PNAS 2020) and that FAN1 nuclease activity is highly sensitive to ionic strength conditions and below detection limits at physiological salt concentration (Phadte et al, PNAS 2023), I would strongly recommend analysing in particular the interplay between FAN1 and MutSb-MutLg in strand-specific CAG repair at near-physiological ionic strength conditions (e.g. >120 mM KCl).

2. Figure 1d and Supplementary Figure 1: From these unspecific nicking assays, the authors conclude that MutLg nuclease activity is differently regulated depending on whether the MutS heterodimeric partner is MutSb or MutSg. To further corroborate these findings and assess whether the striking stimulation of MutLg by MutSb is at least partially dependent on a specific physical interaction between the MSH3 subunit of MutSb (harbouring a MIP motif) and MutLg, the authors should

also test the combination of MutSa and MutLg, as well as a MIP-mutant of MutSb (MSH2-MSH3 F27A/F28A) defective in MLH1 binding (Iyer et al., 2010).

3. Figures 1 e-g (T4 loop substrate): During canonical mismatch repair the MutSb-MutLa (MLH1-PMS2) machinery deals with small and large insertion/deletion loops (IDLs) from 1 up to 15 bp. Moreover, unlike MLH3 but similar to FAN1, PMS2 has been demonstrated to act as an expansion suppressor in HD mouse models (Pinto et al., 2025), most likely because MLH1 is more to partner with MLH3 in the absence of PMS2 and because MutLa, unlike MutLg, introduces nicks randomly in both DNA strands. Thus, to verify the model and significantly strengthen their data, the authors should use the non-nicked pSAM_T4 loop as well as the CAG4 loop substrates and test whether MutSb-MutLa, unlike MutSb-MutLg, indeed makes incisions in both strands. Similarly, experiments shown in Figure 3 (dCTP incorporation and TIDER) could be performed with MutLa as well to check whether modifications of the sequence in both strands can be observed, including loop contractions (like for FAN1), as expected. In fact, processing of the nicked and non-nicked CAG4 loop substrate by MutSb-MutLg has not been as rigorously tested as compared to the T4 loop but mostly used later for TIDER analysis and FAN1 nuclease assays. To me, it is key to compare the cleavage pattern of the CAG4 loop substrate by the two machineries (MutSb-MutLg vs. FAN1) before moving to more complex 'repair assays' including DNA polymerase delta (see also point 5).

4. On page 5 (lanes 119-121), the authors conclude that "EXO1-D173A was moderately inhibitory when combined with MutSb-MutLg (Supplementary Fig. 4a,b), in agreement with its protective function in triplet repeat expansion and our nicking assays (Fig. 1d)." This is an overstatement, as to me cleavage in absence or presence of EXO1 look pretty much identical. First of all, the authors should provide experimental details about how much EXO1-D173A protein is used in all these assays (information is missing in 'Methods'). Second, I suggest performing a titration experiment with increasing concentrations of EXO1-D173A (in presence of high salt concentrations, see above), starting from equimolar amounts of MutSb. Finally, to reveal whether the nuclease-independent inhibition is due to interference of EXO1 with MutSb-MutLg interaction, (similar to what is proposed for FAN1), the authors should analyse the impact of an EXO1-D173A MIP-box mutant (F506A/F507A), displaying reduced interaction with MLH1 (Guan et al., Cancer Cell 2020). In this paper, the authors show that MLH1 binding to EXO1 restrains EXO1 5'-3' exonuclease activity, which could also be true for MutLg-mediated restriction of FAN1 5'-3' exonuclease.

5. As aforementioned (point 3), I think it is critical in Figure 4 to compare, in parallel, the cleaving pattern of unnicked pSam_(CAG)4 substrates by either FAN1 or MutSb-MutLg alone as well as by FAN1 or MutSb-MutLg in presence of PCNA+RFC to obtain further mechanistic insights into overall substrate processing activity by the two different 'machineries'. Moreover, GLOE-Seq analysis, nicely demonstrating predominant FAN1 incision specifically 1 nt downstream of the CAG4 loop (Figure 4d), should be performed with the second machinery consisting of MutSb-MutLg-PCNA-RFC.

6. On page 8/9, the authors state that "We showed that that the nuclease activity of FAN1 removes extrahelical loops, preventing loop recognition by MutS β , which fails to activate MutLg." In my opinion, more experimental data is clearly needed to support this conclusion. In HD brains, expanded CAG repeats in the HTT gene somatically expand further over time in the presence of "physiological levels" of MutSb and FAN1, suggesting that (i) FAN1 must be present at lower levels compared to MutSb but that (ii) increased FAN1 protein levels might be beneficial for slowing down the expansion process, and thus, HD pathogenesis. In Figure 5a, the inhibition of MutSb-MutLg nuclease is not assessed when equimolar amounts of FAN1 and MutSb-MutLg (75 nM) are used but only when present in molar excess of FAN1 (e.g. 50% inhibition when FAN1 is present at 250 nM). Thus, I would recommend the authors to be more specific in the text on p.9 (lanes 270-272). Most importantly, the reciprocal experiment has not been done, investigating whether increasing concentrations of MutSb compete with FAN1-mediated cleavage of oligo-based CAG4 loop DNA substrate.

7. The final experiments shown in Figures 5c-e are exciting but it is again absolutely critical here to describe in the text (and in the methods) the FAN1 concentration used in these 'competition assays'. Is it the same amount as for MutSb and MutLg (75 nM) or has a molar excess for FAN1 been used (e.g. 125 or 250 nM as in Figure 5a)? It is also important to perform experiments with (i) increasing concentration of FAN1-wt, FAN1-nd and FAN1-nd-MIP-MIM mutant (in Figure 5c) as well as (ii) titrating MutSb protein levels (in Figure 5d) to understand at least in these reconstituted reactions how much excess of FAN1 over MutSb is required to suppress the MutSb-MutLg incision pathway, and, vice versa, how much MutSb is required to prevent the FAN1 incision pathway, thus driving CAG expansion. Similarly, in the TIDER analysis (Figure 5e), the authors should indicate the precise protein amounts of MutSb-MutLg and FAN1 used and discuss the results in the context of protein stoichiometry present in the reaction. Finally, there is a clear band running at around 3.1 knt in lanes 4-6 detect by both P4 (4c) and P1 (4d) probes, yet this product is not described anywhere in the text.

Additional points:

- Figure 1d: The first condition tested does not include MutSy but only MutLy. Change graph accordingly.
- Supplementary Figures 1b-c: Representative agarose gel images clearly display the formation of an additional product following incubation of supercoiled covalently-closed DNA with MutSb-MutLg, suggesting the formation of linearized DNA. The authors should explain in the text how such a second product may have been generated by MutSb-MutLg, and label the bands accordingly.
- Figure 3c: The authors should provide an explanation why only 2% of the products contained (CTG)₄ sequence inserted opposite the (CAG)₄ loop in the template strand? What type of sequences are present in the remaining 98% of the products?
- Supplementary Figures 6b-d: Representative agarose gel images clearly display the formation of an additional product following incubation of supercoiled covalently-closed DNA with FAN1 and RFC, suggesting the formation of linearized DNA. The authors should explain in the text how such a second product may have been generated by FAN1 and label the bands accordingly.
- Supplementary Figures 6b-c: The stimulation of FAN1 endonuclease by RFC and PCNA in cleaving unnicked

pSAM_(CAG)₄ has to be validated using proper statistical tests. Moreover, a study from the Pluciennick lab (available on BioRxiv) has shown that stimulation of FAN1 nuclease activity depends on a physical interaction between PCNA and FAN1 and not, as suggested here, on a direct interaction between RFC and FAN1. The authors should comment on this.

- Supplementary Figure 6f: There is a strong (not 'moderate' as described in the text) stimulation of FAN1-mediated incision of the looped strand in presence of RFC alone in absence of PCNA (lane 6). Can this result be recapitulated at physiological salt concentrations?
- Supplementary Figure 7a: The authors should test whether MutSb-MutLg incisions in the non-loop containing strand can also be observed on the oligo-based substrate.
- Figure 5b and Supplementary Figure 7i: It is impossible to judge from these MLH1-IP experiments how much FAN1 is bound to MLH1 in absence or presence of MutSb, as input levels for MBP-FAN1 are missing from the western blots in both experiments. Moreover, it would be highly informative to probe also for MSH3 and MLH3 in these experiments.
- Include an additional panel in Figure 6, depicting how FAN1, through nuclease-dependent and -independent mechanisms, suppresses MutS-MutLg-mediated repeat expansion and instead promotes repeat contractions in non-dividing cells.
- The correct terminology is "FAN1-MIP-MIM" and not "FAN1-MIM-MIP" as the MIM is located C-terminal from the MIP box. Please change the Figures accordingly.
- Typo: P.4, lane 101: (Supplementary Fig. 2 a,b).
- Typo: P.8, lane 228: (compare Figure 2c and Figure 4d)

Version 1:

Reviewer comments:

Reviewer #1

(Remarks to the Author)

The revised MS thoroughly addressed all questions, concerns and suggestion of the three reviewers. Consequently, it is much improved and ready for the publication.

Reviewer #2

(Remarks to the Author)

The authors have addressed all of my concerns as well as the concern of the other reviewers. This is an excellent story that should be published as is. It is of very high quality and rigor and I would like to congratulate the authors on a very nice piece of work.

Reviewer #3

(Remarks to the Author)

I congratulate the authors for doing an excellent job in satisfactorily addressing all comments in the point-to-point response and in providing new experimental data that strengthen their findings.

July 11, 2025

Response to reviewers

Manuscript: NCOMMS-25-21199-T

We would like to thank the reviewers for their interest in our manuscript and their helpful suggestions. Below we list our point-by-point responses to the individual comments.

REVIEWER COMMENTS

Reviewer #1 (Remarks to the Author):

This is an exciting biochemical study that sheds new light on the mechanisms of repeat expansions and contractions. I shall add right away that this is the most clear, beautiful biochemistry that I've seen in years!

The study has two major results. The authors used double-stranded DNA substrates containing a (CAG)₄ extrahelical loop in one of its strands. They convincingly demonstrated that MutS β -MutL γ complex efficiently cleaves the strand opposite to this loop. RFC-loaded PCNA has a moderate inhibitory effect on the efficiency of the cleavage but restricts it to a site 5' to the loop on the opposite strand. This incision site was shown to serve as a primer for DNA synthesis by Pol δ (in conjunction with RPA, RFC and PCNA), which ultimately leads to the insertion of the (CTG)₄ sequence complementary to the extrahelical loop sequence, i.e. repeat expansion equivalent.

The second major result is that FAN1 nuclease - which is known to counteract repeat expansions - cleaves covalently closed DNA 3' to the extrahelical loop followed by exonucleolytic DNA degradation. In the presence of RFC and PCNA, the endonuclease activity of FAN1 was enhanced, while its exonuclease activity was suppressed. When Pol δ was added to the reaction, its exonuclease activity excised the loop, and subsequent resynthesis created DNA strand without the repeat, i.e. repeat contraction.

Altogether, this is a convincing demonstration of repeat expansions in vitro triggered by MutS β -MutL γ complex versus repeat contractions triggered by the FAN1 nuclease.

I have just two comments with regard to biology of the matter. First, in the author's system, an elementary step of repeat expansion is limited to the loop-out size. Translating this observation into humans might explain the late onset of most repeat-expansion diseases, as many consecutive rounds of the loop-out addition would be needed to reach the disease threshold. The authors may want to expand on this point. Second, I very much like the model of larger repeat expansions in proliferating cells (Fig. 6b), which are driven by nicks, subsequent replication fork collapse and DSB repair by HR. Notably, this exact sequel of events was recently demonstrated in the paper of Li et al (PMID: 39585990), which the authors forgot to cite.

REPLY: We thank the reviewer for the enthusiastic support of our manuscript. Regarding the two discussion points:

(1) We now mention more clearly that the "strand displacement" model predicts the expansion to occur stepwise, corresponding to the loop-out size. Indeed, the data fits with studies showing that the expansion in non-replicating cells is slow and gradual (PMID: 39824182; PMID: 39938516).

(2) We have added Ref. PMID: 39585990, thank you for the reminder.

Reviewer #2 (Remarks to the Author):

1. Noteworthy results

The authors provide convincing and detailed mechanistic insight into how three key protein complexes/proteins - MutS β , MutL γ , and FAN1 - may contribute to trinucleotide repeat (TNR) instability

They reproduced some published data by other groups, especially around MutL γ nicking the non-nicked strand and FAN1 nicking the looped strand. They however provide novel insights, arguing that FAN1 does not need a pre-existing nick to cut the looped strand as previously suggested (Phadte et al). Moreover, they uncovered a sequence preference for MutS β -activated MutL γ not previously appreciated. RFC alone or with PCNA restrict cutting by both FAN1 and MutL γ . This was substantiated by a very nice dCas9 experiment showing that it is a physical barrier. Intriguingly, RFC alone could enhance the nuclease activity of FAN1. Importantly, they find that MutS β , MutL γ , RPA, RFC, PCNA, and Pol δ are sufficient to create expansions in vitro. They identified a potential role of FAN1 and Pol δ in creating contractions.

2. Will the work be of significance to the field and related fields? How does it compare to the established literature? If the work is not original, please provide relevant references.

The research presented in this paper will be of great interest in the field, building upon supporting data from years of work in experimental models and genome-wide association studies to explain the oppositional roles for MutL γ and FAN1 at the DNA strand and nucleotide resolution. The work strongly supports ongoing translational research by other groups targeting MSH3 (MutS β), MLH3 (MutL γ), and FAN1 and may inform targeting of new epitopes or interactions.

The data presented also explain previous genetic reports of FAN1 competition with MutL γ at the nuclease and structural levels. The authors greatly expand on the established literature, using very well-controlled experiments and innovative techniques (e.g., GLOE-Seq) to improve understanding of the events leading to MutS β -MutL γ and FAN1-mediated repeat instability, whilst also defining novel roles for additional key proteins.

3. Does the work support the conclusions and claims, or is additional evidence needed?

The work does generally meet the conclusions and claims. I would like to highlight the numerous controls done on how the purified proteins are active on control substrates, for example, and how yeast proteins cannot substitute for human ones. Overall, the experiments are performed at a very high standard.

However, some additional statistical evidence would support the results further:

- Where Southern blots have been quantified and graphed, p-values and significant differences should be included on all graphs (ie. Fig. 1i, Fig. 4c)
- All Southern blots should be quantified, with statistical results detailed in the supplementary data, if not in main text.
- Clarification on how data were statistically analysed – if t-tests were used, were all data individually compared or select data only? If select, then which and why? Would an ANOVA be better in much of these cases?
- A statistical analysis section is missing from the methods

REPLY: We thank Reviewer 2 for raising this point. We have now added quantitative analyses and appropriate statistics for all panels in which we make quantitative claims (Fig. 1i, 2a, 4c, 5e, f, g, h, Supplementary Fig. 4c, Supplementary Fig. 7b, c and d), and inserted a new “Statistical analysis” subsection in the Methods. The specific test used in each panel, including the sample size (n) is mentioned in each legend. Comparisons between two groups were made by unpaired, two-tailed Student’s t-test. Datasets involving more than two groups under a single variable were analyzed by ordinary one-way ANOVA followed by Tukey’s multiple comparisons test to control the family-wise error rate. Experiments with two independent factors (protein components and looped vs. non-looped DNA strand, such as in our FAN1 assay, Fig. 4d) were analyzed by a full two-way ANOVA (interaction term included), followed by Tukey’s multiple-comparisons tests to examine simple effects within each row and column, while controlling the family-wise error rate.

4. Are there any flaws in the data analysis, interpretation and conclusions? - Do these prohibit publication or require revision?

Beyond the clarification on the statistical analysis above, the manuscript can be published as is. I have, however, some minor comments that would be good to address.

It would be good to specify why only CAG loops were used and not CTG ones. I would not want to have the authors redo the whole of the experiments with CTG loops as they have not made claims about CTG loops and it would double the size of the paper.

REPLY: We thank Reviewer 2 for this suggestion. Indeed, we performed preliminary experiments using also (CTG)₄ and (CTG)₈ loops and observed no significant differences depending on the sequence of the loop-out (Fig. R1). Our future work will focus on how loop size, sequence and secondary structure influence cleavage efficiency, as well as how various protein co-factors (RPA, PCNA, etc.) affect the reactions depending on the loop size/sequence/structure, and we prefer to present those detailed results comprehensively in a future manuscript.

Figure R1. Comparison of DNA cleavage by MutSβ and MutLγ of DNA substrates with various loop-out sequences and lengths.

How were experimental conditions (e.g., protein concentration and buffer conditions) decided upon? Were these physiological conditions or were there trials of different conditions?

REPLY: The reaction conditions were optimized based on our nicking assays, which allow higher throughput than the labor-intensive Southern blots. In this way, we chose protein concentrations that consistently yield reliable cleavage, but that are not saturated. With respect to buffer conditions, and particularly ionic strength (salt concentration), we note that the activity of MutL γ and MutS β is inhibited by salt (above ≥ 75 mM NaCl, see Fig. R6), in agreement with our report and the work from the Hunter lab on the role of MutS γ -MutL γ (PMID: 32814904, 32814343). Therefore, we needed to maintain NaCl at 25-45 mM range throughout all experiments, as specified in Methods.

i. Supplementary Figure 1. However, can MutS β stimulate MutL γ DNA cleavage at DNA substrates with CAG loops? As this would have an impact on the necessity of MutS β (and the therapeutic targeting of MSH3) in MutL γ -mediated loop incorporation and repeat expansion.

REPLY: Our manuscript focuses on the interplay of MutS β and MutL γ , so we assume the reviewer was referring to either MutS β with MutL α , or MutS α with MutL γ . Regarding the first pair, and also in response to reviewer #3, we added several supplementary data panels (Supplementary Fig. 3f, g, h, i) showing that the cleavage is targeting randomly both strands, likely resulting from random loading of PCNA (on both strands) at the loop sites. The relevance of these data is discussed. Regarding MutS α with MutL γ , we note that MutS α (MSH2-MSH6) cannot recognize a loop-out of the tested length (CAG)₄, and therefore would not activate MutL γ on this substrate (see also Fig. R2). Additionally, even in the non-specific nicking assays (Supplementary Fig. 1a), we never observed a stimulation of MutL γ by MutS α , suggesting that there is likely no interplay between the two factors, irrespective of the DNA substrate.

Figure R2. Comparison of DNA cleavage by the two pairs MutS α -MutL γ and MutS β -MutL γ on pSam (CAG)₄

ii. This is beyond the scope of the study and only for the authors to think about. One outstanding question from the genetics is that all three MutL complexes are essential for repeat expansion. How they would interact and indeed whether they process this loop out substrate separately would be a very interesting question to answer using the approaches herein.

REPLY: We appreciate the reviewer's insightful comment. We are currently investigating MutL β using this experimental setup and hope these studies will yield mechanistic insights.

iii. Figure 2a: There is a distinct band in lane 4 (without MutS/MutL) which is only present in the other lanes with MutS and MutL. Is this cleaved DNA?

REPLY: This band, which sometimes appears just below the substrate, results from plasmid DNA that remains uncleaved by ScaI before Southern blotting. We have added "*" to mark this band in the panels where this appears and explain it in the legends.

iv. "RFC directly controls FAN1, which is likely dependent on a direct interaction" (Line 191 – 192) Can the authors show direct interaction between RFC and FAN1 using pulldown or other method? Or speculate how

RFC may be interacting with FAN1 in the absence of PCNA? It is difficult to reconcile how RFC may regulate FAN1 in the absence of PCNA. Specifically, what do they mean by ‘loading FAN1’ here?

REPLY: We appreciate the suggestion and have added a new panel in Fig. 4 (panel a) showing that human RFC co-immunoprecipitates with FAN1, in agreement with the Predictomes study (PMID: 38645019). Additionally, RFC is well known to bind DNA, and e.g. in previous work by Modrich and colleagues was implicated to limit EXO1 activity starting at DNA nicks (PMID: 15225546), independently of PCNA. Thus, RFC’s regulation of FAN1 likely involves both direct protein–protein contacts and DNA binding. This point was made clearer in the revised text.

v. PCNA only modestly increases FAN1 endonuclease activity when added to RFC (Line 189 – 194) If RFC stimulates FAN1 endonuclease activity, and addition of PCNA only modestly increased this further, could the authors speculate on discrepancies from other data? Namely the necessity for both RFC and PCNA for DNA cleavage of loop-containing DNA substrate (Phadte et al., 2023) and the impact of the FAN1-PCNA interacting p.R507H onset-hastening variant (Li et al., 2024).

REPLY: We thank the reviewer for requesting clarification. We have modified the text to make it clearer that RFC primarily limits the exonuclease activity of FAN1 (independently of PCNA), and only a weak stimulation of FAN1 endonuclease was observed on plasmid-based substrates by RFC. In contrast, PCNA, loaded by RFC, strongly stimulates the FAN1’s endonuclease function and does not affect the exonuclease activity. There is no fundamental discrepancy with Phadte et al., 2023, we observed the same effects; however, we extend their data in two key aspects: (1) the direct interplay of RFC and FAN1 (supported by physical interaction reported in the revised manuscript); and (2) we note that a preexisting nick is not strictly necessary for the effect of RFC-PCNA on FAN1, i.e. that PCNA can get loaded on the DNA substrate at the loop sites (PMID: 23610416).

With regard to the FAN1 R507H mutant, we expressed and purified the variant and performed nuclease assays using the CAG-loop-containing oligonucleotides (Fig. R3, right). In accord with the data of Li et al., 2024, under low-salt conditions (45 mM NaCl), the R507H protein and the wild-type enzyme exhibited comparable activity in nuclease assays, and co-immunoprecipitation experiments conducted in the absence of DNA showed that the R507H mutation alone does not reduce PCNA binding (Fig. R4). These findings are consistent with those of Li et al., 2024, who reported that the R507H mutant is impaired in forming the FAN1–PCNA–DNA complex, and effects of the mutation in nuclease assays were only evident at high salt concentration. Due to our focus on comparing activities of FAN1 and MutSβ–MutLy, and the need to keep the salt concentration lower to allow MutLy to be active, we did not perform the experiments in high salt, also considering that this was already done (Li et al., 2024). We assume that there are additional contact points between FAN1 and PCNA, and the single point mutant does not disrupt it under our experimental conditions.

Figure R3. Left, Polyacrylamide gels of FAN1-R507H stained with Coomassie Brilliant blue. Right, DNA cleavage activity of WT FAN1 versus the R507H variant +/- RFC-PCNA on an oligonucleotide DNA substrate containing (CAG)₄

Figure R4. Pull down assay comparing MBP-tagged WT FAN1 and MBP-tagged FAN1 R507H. Each immobilized protein was incubated with RFC alone and PCNA alone.

vi. Were human and yeast protein interaction experiments conducted under the same conditions? As yeast and human proteins have been reported to be affected differently by different cations (Kadyrova et al., 2020).

Where possible, referring to specific lanes in a figure would be very helpful when making statements in the text. The authors have done this sometimes and I found it to make reading the paper much more straightforward.

REPLY: We appreciate this important point. All assays with the combinations of yeast and human proteins presented in this manuscript were performed at 37 °C, under the same conditions. We now include an important control (Supplementary Fig. 7e,f) showing that the yeast proteins *per se* are active, both at 30 °C and 37 °C, with 25 mM NaCl and 5 mM Mg²⁺, matching the parameters used for the assays with human proteins. The readout for the activity of the yeast proteins was strand displacement synthesis, well known to be stimulated by yeast Pif1 (PMID 24025768, 33823531). We agree on using lane numbers in the description of the data - this is now done in multiple cases.

vii. Additional clarity in distinguishing between extrahelical loops and CAG4 loops would be useful. I initially thought that CAG4 extrahelical loops were used throughout the experiment, but I think the initial experiments using extrahelical loops were non-CAG loops? Some additional distinction in the text (to mirror the differences in the figures) would be helpful.

REPLY: Thank you, this point was made clearer in the Figures and Legends.

Thank you for the comments below!

Other minor issues:

- Line 121: “repat” instead of repeat. **Done**
- Line 101: no figure referenced **We included also there**
- Figure 1F: no Y-axis scale. Same scale as E? **Scale was added**
- Figure 1J: DNA ladder label overlapping with blot. **Done**
- Figures 1J and 1K: P2 and P1 blots/maps are in different order to 1E and 1F. **The order follow the story telling, so we decided to let it as it was.**
- 1J: no red triangle? **There was no red triangle (now green triangle) there because the red triangle was a result of the accumulation of MutLy when Cas9 blocks its progression, as described in the legend.**
- Figure 1L: red triangle difficult to distinguish from orange – I suggest changing the colour scheme. **Changed into green.**
- Figure 4 has a huge amount of data. Could this be split into two figures or sectioned off? **We divided Fig. 4 into two separate Figures.**
- Figure 4C: changing “P4” to black would make interpreting strand preference easier. **Done**

- Figure 4J: dividing actual blot would be helpful, rather than dashed line below. **Thank you, we prefer to keep it as a same blot to demonstrate that the product distributions match the various setups, our point was made clearer in the text.**
- Figure 4K figure legend: states “rectangle” rather than triangle. **Done.**
- Figure 5C: bands are quite difficult to make out, would be helpful to quantify and present.
- Arrow colour difficult to distinguish. **See reply to first comment**
- Figure 6: Model doesn't include FAN1. **FAN1 was included in the final model.**
- Supplementary 3B: one substrate map missing **Done.**

7. Is the methodology sound? Does the work meet the expected standards in your field?

Yes, the methodology is sound and meets expected standards.

8. Is there enough detail provided in the methods for the work to be reproduced?

The methods are very detailed. An addition of how statistical analyses were done should be included.

Reviewer #3 (Remarks to the Author):

In this manuscript, Cejka and colleagues employ state-of-the art protein biochemistry and combine elegant in vitro assays with genome-wide next generation sequencing methods (GLOE-Seq and TIDER analysis) to evaluate the functional interplay between two DNA repair machineries, driven by either MutLg (MLH1-MLH3) or FAN1 nuclease activities, on the fate of CAG repeat stability. Most notably, their results, highlighting competing mechanisms which either facilitate (MutLg) or inhibit (FAN1) CAG repeat expansion, are largely consistent with previous biochemical studies using purified proteins as well as cell extracts (e.g. Kadyrova et al., 2020; Phadte et al. 2023) and with recent genetic data obtained in mouse models (e.g. Mouro Pinto et al. 2024).

Their experimental work is mostly solid and overall, of high technical quality. In some instances, however, and as outlined in detailed below, additional evidence is required to support or corroborate their conclusions. There are also important information missing regarding methodology of the in vitro reconstituted reactions, most notably the salt concentration (it looks as if all reactions were carried out at non-physiological ionic strength) and FAN1 vs. MutSb (MSH2-MSH3) concentration (i.e. stoichiometry is a key determinant of DNA repair outcome). Otherwise, their findings presented in Figure 5 rather contradict the mechanism underlying HD pathogenesis, namely that due to higher abundance of MutSb and/or higher binding affinity of MutSb to extrahelical loops compared to FAN1 (which is commonly expressed at very low levels), CAG repeats tend to expand with time in HD brains (with both MMR and FAN1 pathways present).

In conclusion, this study not only enhances our mechanistic understanding of somatic repeat expansion, which is highly relevant for the HD field, but is also of broad interest to the entire DNA repair community. Thus, should the authors be able to appropriately address the comments, a significantly revised version of manuscript would certainly fit the scope of Nature Communications.

Major points:

1. According to the Methods description, none of the reaction buffers in the reconstituted biochemical assays do contain any monovalent chloride salts (KCl or NaCl). This raises the important question as to whether certain specific stimulatory (e.g. by RFC-PCNA on FAN1 nuclease activity) or inhibitory mechanisms (e.g. by RFC-PCNA on MutSb-MutLg) observed here only occur at non-physiological (i.e. low salt) conditions or whether they are also active at physiological ionic strength? Considering the fact that MutLg displays DNA

binding activity in absence of MutSb complexes at low salt conditions (Kadyrova et al, PNAS 2020) and that FAN1 nuclease activity is highly sensitive to ionic strength conditions and below detection limits at physiological salt concentration (Phadte et al, PNAS 2023), I would strongly recommend analysing in particular the interplay between FAN1 and MutSb-MutLγ in strand-specific CAG repair at near-physiological ionic strength conditions (e.g. >120 mM KCl).

REPLY: We recognize the importance of this comment. The salt concentrations in our nuclease assays were generally very low, most of the time corresponding to what was introduced with the protein storage buffer (25-45 mM), and compensated in individual lanes to make them exactly the same in all lanes. This point was added to Methods. The reason for that is MutLγ, which is even more sensitive to salt than FAN1. This is also the case together when MutLγ is together with MutSβ, and in agreement with our report and the work from the Hunter lab on the role of MutSγ-MutLγ (PMID: 32814904, 32814343). Whatever we tried (and we tried a lot), we could never get MutLγ to be active at physiological salt. The stimulatory or inhibitory effects were consistent when we went higher with salt, but the overall activities went down (not shown). While RFC-PCNA activate FAN1 at higher salt - we do see the same as Phadte et al, PNAS 2023 - the main angle of our assays was to observe MutLγ, its co-factors and competing activities, so we needed to select conditions when MutLγ is active.

2. Figure 1d and Supplementary Figure 1: From these unspecific nicking assays, the authors conclude that MutLγ nuclease activity is differently regulated depending on whether the MutS heterodimeric partner is MutSb or MutSγ. To further corroborate these findings and assess whether the striking stimulation of MutLγ by MutSb is at least partially dependent on a specific physical interaction between the MSH3 subunit of MutSb (harbouring a MIP motif) and MutLγ, the authors should also test the combination of MutSa and MutLγ, as well as a MIP-mutant of MutSb (MSH2-MSH3 F27A/F28A) defective in MLH1 binding (Iyer et al, 2010).

REPLY: We note that the combinations of MutSα and MutLγ do not result in any activity, see Supplementary Fig. 1a and Fig. R2. Hence, the interplay of MutSβ and MutLγ is highly specific.

To assess the impact of the MLH1-binding-defective MutSβ variant, we purified the mutant (MSH2-MSH3 F27A/F28A) (Fig. R5) and evaluated its ability to stimulate MLH1-MLH3. Contrary to expectations, this mutation did not impair the activation of MLH1-MLH3 (Fig. R5), not even in the presence of higher NaCl concentration (50 mM; 75 mM and 100 mM NaCl). Because the original study by Iyer et al. 2010 focused on MutSβ's interaction with MutLα and PCNA, other contact points with MutLγ likely exist, possibly explaining why the motif does not impact the functional interaction with MLH1-MLH3.

Figure R5. Left, polyacrylamide gels of MutSβ F27A/F28A stained with Coomassie Brilliant blue. Right, representative nicking assay with pSam scDNA and the indicated proteins in presence of 7 mM NaCl coming from the protein storage buffer, 50 mM NaCl, 75 mM NaCl or 100 mM NaCl.

3. Figures 1 e-g (T4 loop substrate): During canonical mismatch repair the MutSb-MutLa (MLH1-PMS2) machinery deals with small and large insertion/deletion loops (IDLs) from 1 up to 15 bp. Moreover, unlike MLH3 but similar to FAN1, PMS2 has been demonstrated to act as an expansion suppressor in HD mouse models (Pinto et al., 2025), most likely because MLH1 is more to partner with MLH3 in the absence of PMS2 and because MutLa, unlike MutLg, introduces nicks randomly in both DNA strands. Thus, to verify the model and significantly strengthen their data, the authors should use the non-nicked pSAM_T4 loop as well as the CAG4 loop substrates and test whether MutSb-MutLa, unlike MutSb-MutLg, indeed makes incisions in both strands. Similarly, experiments shown in Figure 3 (dCTP incorporation and Tider) could be performed with MutLa as well to check whether modifications of the sequence in both strands can be observed, including loop contractions (like for FAN1), as expected. In fact, processing of the nicked and non-nicked CAG4 loop substrate by MutSb-MutLg has not been as rigorously tested as compared to the T4 loop but mostly used later for TIDER analysis and FAN1 nuclease assays. To me, it is key to compare the cleavage pattern of the CAG4 loop substrate by the two machineries (MutSb-MutLg vs. FAN1) before moving to more complex 'repair assays' including DNA polymerase delta (see also point 5).

REPLY: We thank Reviewer 3 for these insightful comments. We performed nuclease assays, analyzed by Southern blotting, using MutL α in conjunction with MutS β on both (T)₄ and (CAG)₄ substrates, with MutL γ and MutS β as controls. As shown in new Supplementary Fig. 3f-i, MutL α indeed cleaves both strands, as opposed to MutS β -MutL γ that cleave almost exclusively the opposite (non-looped) strand.

Moreover, at equimolar concentrations, MutS β -MutL γ mediated cleavage was decreased upon adding MutL α , demonstrating that MutL α had a more "dominant" effect, which agrees with Pinto et al., 2025. We did not follow up with sequencing analysis in order not to further increase the complexity of the data.

4. On page 5 (lanes 119-121), the authors conclude that "EXO1-D173A was moderately inhibitory when combined with MutSb-MutLg (Supplementary Fig. 4a,b), in agreement with its protective function in triplet repeat expansion and our nicking assays (Fig. 1d)." This is an overstatement, as to me cleavage in absence or presence of EXO1 look pretty much identical. First of all, the authors should provide experimental details about how much EXO1-D173A protein is used in all these assays (information is missing in 'Methods'). Second, I suggest performing a titration experiment with increasing concentrations of EXO1-D173A (in presence of high salt concentrations, see above), starting from equimolar amounts of MutSb. Finally, to reveal whether the nuclease-independent inhibition is due to interference of EXO1 with MutSb-MutLg interaction, (similar to what is proposed for FAN1), the authors should analyse the impact of an EXO1-D173A MIP-box mutant (F506A/F507A), displaying reduced interaction with MLH1 (Guan et al., Cancer Cell 2020). In this paper, the authors show that MLH1 binding to EXO1 restrains EXO1 5'-3' exonuclease activity, which could also be true for MutLg-mediated restriction of FAN1 5'-3' exonuclease.

REPLY: We thank Reviewer 3 for these comments and for highlighting the missing information, which we have now included in Methods or Legends. We have observed the inhibition of MutS β -MutL γ by EXO1-D173A in the non-specific nicking assays with negatively supercoiled DNA that is statistically significant (Supplementary Fig. 1a). In Southern blots, using substrates without a nick (relaxed DNA with CAG loop-out), the effect was indeed small, and restricted only to the more distal incisions sites (Supplementary Fig. 4a). However, when nicked DNA was used, the inhibitory effect of EXO1 was more prominent (Supplementary Fig. 4b). We believe that EXO1 binds to the nick site (as shown by Alani, PMID: 37079643) and prevents MutL γ -MutS β sliding beyond that site, inhibiting cleavage downstream, similarly to PCNA.

To investigate the mechanism behind the inhibition of MutL γ -MutS β by EXO1, we used nicking assays and several EXO1 mutants: nuclease-dead (D173A) and nuclease dead in combination with MIP (F506A/F507A) and I403E (impaired in interaction with MLH1) and nuclease dead in combination with KDKD (K185D/K237D, impaired in interaction with DNA) (PMID: 40319035). EXO1 D173A-MIP-I403E mutant inhibits MutL γ -MutS β cleavage comparably to EXO1 D173A, whereas EXO1 D173A-KDKD mutant was less inhibitory (new Supplementary Fig. 4c). Together, these results indicate that the inhibition of MutL γ -MutS β by EXO1 is dependent primarily on its DNA binding activity, rather than on the interaction with MLH1. Such effect may be entirely

non-specific; however, we point out the critical function of EXO1 in the stimulation of MutSy-MutLy (PMID: 40319035), and the MutS homologue (MutSy vs. MutS β) thus determines whether EXO1 is stimulatory. We discuss the data in this context.

5. As aforementioned (point 3), I think it is critical in Figure 4 to compare, in parallel, the cleaving pattern of unnicked pSam_(CAG)₄ substrates by either FAN1 or MutSb-MutLg alone as well as by FAN1 or MutSb-MutLg in presence of PCNA+RFC to obtain further mechanistic insights into overall substrate processing activity by the two different 'machineries. Moreover, GLOE-Seq analysis, nicely demonstrating predominant FAN1 incision specifically 1 nt downstream of the CAG₄ loop (Figure 4d), should be performed with the second machinery consisting of MutSb-MutLg-PCNA-RFC.

REPLY: The cleavage of pSam_(CAG)₄ by FAN1 +/- RFC/PCNA is documented in Fig. 4b and 4c, quantitated in panel 4d, with the accompanying GLOE-Seq data (Fig. 4d, with RFC-PCNA) and (Supplementary Fig. 7i, without RFC-PCNA). All datasets show the marked effects of RFC-PCNA. The cleavage of pSam_(CAG)₄ by MutLy, MutS β +/- RFC/PCNA is documented in Fig. 6c and 6d, now complemented with new GLOE-Seq data (Supplementary Fig. 7j) as suggested by the reviewer. We observed that the processing by MutLy, MutS β +/- RFC/PCNA is very similar when looking at (CAG)₄ and (T)₄ loops (Fig. 1g, 2c, Supplementary Fig. 4a, b, Fig. 6c,d). Fig. 6c and 6d also directly compare the effects of MutS β -MutLy vs. FAN1 with RFC-PCNA (i.e. the more physiological condition). We do not have, on the same gel, the comparison MutS β -MutLy vs. FAN1 without RFC-PCNA, however, as this has been documented in multiple previous panels individually, with different substrates, we felt adding that panel would lead to duplication of data, and we hope the reviewer agrees.

6. On page 8/9, the authors state that "We showed that that the nuclease activity of FAN1 removes extrahelical loops, preventing loop recognition by MutS β , which fails to activate MutLg." In my opinion, more experimental data is clearly needed to support this conclusion. In HD brains, expanded CAG repeats in the HTT gene somatically expand further over time in the presence of "physiological levels " of MutSb and FAN1, suggesting that (i) FAN1 must be present at lower levels compared to MutSb but that (ii) increased FAN1 protein levels might be beneficial for slowing down the expansion process, and thus, HD pathogenesis. In Figure 5a, the inhibition of MutSb-MutLg nuclease is not assessed when equimolar amounts of FAN1 and MutSb-MutLg (75 nM) are used but only when present in molar excess of FAN1 (e.g. 50% inhibition when FAN1 is present at 250 nM). Thus, I would recommend the authors to be more specific in the text on p.9 (lanes 270-272). Most importantly, the reciprocal experiment has not been done, investigating whether increasing concentrations of MutSb compete with FAN1-mediated cleavage of oligo-based CAG₄ loop DNA substrate.

REPLY: To substantiate these findings, we performed electrophoretic mobility shift assays with MutS β and various oligonucleotide-based structures that represent the substrate (looped DNA), intermediate (looped DNA nicked by FAN1), nicked DNA (looped DNA cleaved by FAN1 and Pol δ), as well as duplex DNA as a control. We observed that looped and nicked looped DNA are well recognized by MutS β while the subsequent loop removal (nicked DNA) is not a good substrate, in agreement with our model (Fig. 5e).

In addition, we show that a plasmid substrate with a nick (i.e., a substrate likely resulting from the processing of looped DNA by FAN1 and Pol δ) does not activate MutS β -MutLy (Supplementary Fig. 5c and d).

Regarding the reciprocal experiment, i.e. whether MutS β affects FAN1, we have observed, to our initial surprise, that MutS β promotes FAN1 (data not shown). While these results are contrary to expectations with regard to trinucleotide repeat metabolism, we noted a previous publication (PMID: 34228493), suggesting that FAN1 may have a residual function in MMR. In the latter scenario, a stimulation by MutS β would make sense, as FAN1 might function instead of MutL α . As these data are difficult to interpret without substantial further experimentation, we opted not to include the dataset in the manuscript and will focus on it in the future.

7. The final experiments shown in Figures 5c-e are exciting but it is again absolutely critical here to describe in the text (and in the methods) the FAN1 concentration used in these ‘competition assays. Is it the same amount as for MutSb and MutLg (75 nM) or has a molar excess for FAN1 been used (e.g. 125 or 250 nM as in Figure 5a)? It is also important to perform experiments with (i) increasing concentration of FAN1-wt, FAN1-nd and FAN1-nd-MIP-MIM mutant (in Figure 5c) as well as (ii) titrating MutSb protein levels (in Figure 5d) to understand at least in these reconstituted reactions how much excess of FAN1 over MutSb is required to suppress the MutSb-MutLg incision pathway, and, vice versa, how much MutSb is required to prevent the FAN1 incision pathway, thus driving CAG expansion. Similarly, in the TIDER analysis (Figure 5e), the authors should indicate the precise protein amounts of MutSb-MutLg and FAN1 used and discuss the results in the context of protein stoichiometry present in the reaction. Finally, there is a clear band running at around 3.1 knt in lanes 4-6 detect by both P4 (4c) ad P1 (4d) probes, yet this product is not described anywhere in the text.

REPLY: We apologize for the omission of the concentrations, and thank the reviewer for spotting that - everything is now indicated in the Methods or Legends. Please see our comment above about the effect of MutSβ on FAN1.

In our nuclease–Southern blot competition assays below, we observed that at 100 nM FAN1 (compared to 75 nM MutLy), the secondary (more distant to the loop, black triangle) cleavage product is markedly reduced when using the FAN1-ND variant but remains unchanged with the FAN1-ND–MIP–MIM mutant, which only begins to inhibit at 200 nM; at 200 nM the FAN1-ND cleavage beyond the primary cut is completely abolished whereas it partially persists for FAN1-ND–MIP–MIM (Fig. R6). Therefore, this inhibition cannot be attributed only to MLH1 disruption, but additionally to FAN1’s strong DNA-binding activity, which appears to block MutLy nuclease progression along the duplex (as observed for PCNA, EXO1 or dCas9). As we are working with purified recombinant proteins of undefined specific activities, we are careful not to overinterpret the data with respect to the relative concentrations used.

Regarding the band which sometimes appears below the substrate, it results from plasmid DNA that remains uncleaved by Scal and therefore migrates as a single circular molecule - we added a note to legends where this band is visible.

Figure R6. Representative nuclease assay done on pSam (CAG)₄ with the indicated proteins and analyses by Southern blot with the indicated probes.

Same but with increased signal:

Additional points:

- Figure 1d: The first condition tested does not include MutSy but only MutLy. Change graph accordingly. **Done.**

- Supplementary Figures 1b-c: Representative agarose gel images clearly display the formation of an additional product following incubation of supercoiled covalently-closed DNA with MutSb-MutLg, suggesting the formation of linearized DNA. The authors should explain in the text how such a second product may have been generated by MutSb-MutLg, and label the bands accordingly.

REPLY: The observed bands correspond to the linearized product. The substrate used is a negatively supercoiled plasmid DNA, and if MutLy cleaves each strand independently, it can introduce double-strand breaks, as demonstrated by Rogacheva et al. (PMID: 24403070). We have now updated the labels accordingly and clarified this in the main text.

- Figure 3c: The authors should provide an explanation why only 2% of the products contained (CTG)₄ sequence inserted opposite the (CAG)₄ loop in the template strand? What type of sequences are present in the remaining 98% of the products?

REPLY: We thank the reviewer for this comment. Following strand-displacement synthesis by Pol δ , the displaced flap remained attached, and either blocked further progression of strand displacement or “masked” detection of the newly synthesized strand during sequencing. By adding a flap-removing enzyme DNA2 (available in the laboratory), we increased the CTG insertion frequency to 15.5%. We include these data in Supplementary Fig. 6b.

- Supplementary Figures 6b-d: Representative agarose gel images clearly display the formation of an additional product following incubation of supercoiled covalently-closed DNA with FAN1 and RFC, suggesting the formation of linearized DNA. The authors should explain in the text how such a second product may have been generated by FAN1 and label the bands accordingly.

REPLY: As shown in Figure 4d using GLOE seq, FAN1 cleaves primarily at the loop in the presence of PCNA, but this activity is not strictly limited to that site and a small amount of cleavage also occurs opposite the loop. Although minimal, this cleavage generates linearized DNA, giving rise to the observed “extra band”. We have updated the gel labels accordingly and clarified this in the main text.

- Supplementary Figures 6b-c: The stimulation of FAN1 endonuclease by RFC and PCNA in cleaving unnicked pSAM_(CAG)₄ has to be validated using proper statistical tests. Moreover, a study from the Pluciennick lab (available on BioRxiv) has shown that stimulation of FAN1 nuclease activity depends on a physical interaction between PCNA and FAN1 and not, as suggested here, on a direct interaction between RFC and FAN1. The authors should comment on this.

REPLY: We thank the reviewer for this important comment. We have now validated the assays by performing a one-way analysis of variance (ANOVA), and the resulting statistical support is included for Supplementary figures 7b, c, and d.

- Supplementary Figure 6f: There is a strong (not ‘moderate’ as described in the text) stimulation of FAN1-mediated incision of the looped strand in presence of RFC alone in absence of PCNA (lane 6). Can this result be recapitulated at physiological salt concentrations?

REPLY: We do see the stimulation at 44 mM salt, 60 mM salt, but not at 120 mM salt. At the high salt concentration, neither PCNA nor RFC promote FAN1 under our conditions (see Fig. R7).

Figure R7. Representative nuclease assay done on pSam (CAG)₄ with the indicated proteins at indicated salt concentrations.

- Supplementary Figure 7a: The authors should test whether MutSb-MutLg incisions in the non-loop containing strand can also be observed on the oligo-based substrate.

REPLY: Unfortunately, MutLγ and MutSβ do not function on oligonucleotide-based substrates, most likely because the oligonucleotides are too short (we tested up to 82 bp in length), in accord with canonical MMR reactions.

- Figure 5b and Supplementary Figure 7i: It is impossible to judge from these MLH1-IP experiments how much FAN1 is bound to MLH1 in absence or presence of MutSb, as input levels for MBP-FAN1 are missing from the western blots in both experiments. Moreover, it would be highly informative to probe also for MSH3 and MLH3 in these experiments.

REPLY: We have added the input of the MBP-FAN1 variants to Fig. 6b (formerly 5b). We provide here the complete pull-down experiment shown in Supplementary Fig. 8i (Fig. R8). We have not observed a decrease of FAN1 binding to MLH1 in the presence of MutSβ.

Figure R8. Pull down assay. MutLγ (MLH1-MLH3) was immobilized using anti-MLH1 antibody, followed by incubation with MutSβ (MSH2-MSH3) and/or MBP-FAN1 variants, as indicated.

- Include an additional panel in Figure 6, depicting how FAN1, through nuclease-dependent and -independent mechanisms, suppresses MutS-MutLg-mediated repeat expansion and instead promotes repeat contractions in non-dividing cells.

- The correct terminology is “FAN1-MIP-MIM” and not “FAN1-MIM-MIP” as the MIM is located C-terminal from the MIP box. Please change the Figures accordingly. **Done - thank you.**
- Typo: P.4, lane 101: (Supplementary Fig. 2 a,b). **Done.**
- Typo: P.8, lane 228: (compare Figure 2c and Figure 4d) **Done: the new comparison is now between Fig. 2c and Supplementary Fig. 7j.**

August 22, 2025

Response to reviewers

Manuscript: NCOMMS-25-21199-T

There were no additional comments or requests from the reviewers.